# Why multilingual, and how to keep it—An evolutionary dynamics perspective

Zhijun Wu *

Department of Mathematics, Iowa State University, Ames, Iowa, United States of America

* zhijun@iastate.edu

## Abstract

While many languages are in danger of extinction worldwide, multilingualism is being adopted for communication among different language groups, and is playing a unique role in preserving language and cultural diversities. How multilingualism is developed and maintained therefore becomes an important interdisciplinary research subject for understanding complex social changes of modern-day societies. In this paper, a mixed population of multilingual speakers and bilingual speakers in particular is considered, with multilingual defined broadly as zero, limited, or full uses of multiple languages or dialects, and an evolutionary dynamic model for its development and evolution is proposed. The model consists of two different parts, formulated as two different evolutionary games, respectively. The first part accounts for the selection of languages based on the competition for population and social or economic preferences. The second part relates to circumstances when the selection of languages is altered, for better or worse, by forces other than competition such as public policies, education, or family influences. By combining competition with intervention, the paper shows how multilingualism may evolve under these two different sources of influences. It shows in particular that by choosing appropriate interventional strategies, the stable co-existence of languages, especially in multilingual forms, is possible, and extinction can be prevented. This is in contrast with major predictions from previous studies that the co-existence of languages is unstable in general, and one language will eventually dominate while all others will become extinct.

## Introduction

As the world becomes increasingly globalized, more people become multilingual across the continents. It is reported that now more than half of the world's population speak at least two languages [1]. While many languages are in danger of extinction, multilingualism is being adopted as a common way of communication among different language groups, and is playing a unique role in preserving language and cultural diversities [2]. Therefore, how multilingualism is developed and maintained becomes an important research subject in linguistics, social studies, and beyond [3]. The promotion and protection of multilingualism has also been a hot topic long discussed in public [4].

**Citation:** Wu Z (2020) Why multilingual, and how to keep it—An evolutionary dynamics perspective. PLoS ONE 15(11): e0241980. https://doi.org/10.1371/journal.pone.0241980

**Data Availability Statement:** All relevant data are within the manuscript.

**Funding:** ZW receives support from the Simons Foundation on this work through the Mathematics and Physical Sciences Collaboration Grants for Mathematicians (Award Number: 586065). The

funders had no role in study design, data collection and analysis, decision to publish, or preparation of the manuscript.

**Competing interests:** The authors have declared that no competing interests exist.

Languages compete and spread among their speakers, as genes are inherited and passed down to biological populations, where some are selected while others become extinct [5]. Genes may be carried over in mixed forms. So are languages by multilingual speakers. Genes flow in and out of a population and are subject to constant mutations among their variants. Speakers also migrate and can as well be converted from speaking one language to another. In this paper, a mixed population of multilingual speakers and bilingual speakers in particular is considered, with multilingual defined broadly as zero, limited, or full uses of multiple languages or dialects, and an evolutionary dynamic model for its evolution is proposed, similar to that for genetic evolution [6].

Unlike genetic evolution though, the uses of languages are not only dependent of competition, but also subject to various societal interventions, common in social or cultural evolution. The proposed model consists of two different parts accordingly, formulated as two different evolutionary games, respectively. The first part accounts for the selection of languages based on the competition for population and social or economic preferences. It assumes that the speakers are able to adjust their ways of using single or multiple languages, although in reality, it may take time and extra learning efforts. The second part relates to circumstances when the selection of languages is altered, for better or worse, by forces other than competition such as public policies, education, or family influences. Under these circumstances, the number of speakers of one language may be increased or decreased due to the impact of certain public policies or the migration of the speakers in or out of the population, etc.

Here are a few examples where both language competition and societal intervention may play a role for the development and maintaining of multilingualism: In twenty-five European countries surveyed, 56 percent of the population speak at least two languages, and 28 percent speak at least three languages [1]. In the United States, over 20 percent of the US population are bilingual [7], and as high as 70 percent of immigrant families speak a language other than English at home [8, 9]. In China, almost all different provinces have their own local Chinese dialects, yet by 2015, over 73 percent of the population have learned Mandarin Chinese, a once northern Chinese dialect later promoted as the official Chinese by the government [10]. However, Cantonese is better preserved in Canton than all other Chinese dialects, as the locals have kept many newspapers, radio broadcastings, movies and TV series in Cantonese [11]. Hebrew, an extinct ancient language, revived in 19th century and later became one of the official languages of Israel, with 9 million speakers worldwide today [12].

Much work has been done on modeling language competition, although not specifically for the evolution of multilingualism. A well known model was proposed by Abrams and Strogatz in 2003 for the study of language death [13]. The model was later extended to more general and complex cases by several other groups [14–17]. The models along this line focus mainly on language competition for population and social or economic preferences. In general, the models predict that one language will eventually dominate the population while all others become extinct, and the co-existence of languages is unstable and hard to sustain [13, 16, 17]. While successfully applied to some language populations, these models have not explicitly distinguished language competition from possible societal interventions that may reverse the course of language changes. By combining language competition with possible societal interventions, this paper shows how multilingualism may evolve under these two different sources of influences. It shows in particular that by choosing appropriate interventional strategies, the stable co-existence of languages, especially in multilingual forms, is possible, and extinction can be prevented, as seen in many multilingual communities.

The paper has three core sections titled Evolutionary Models, Dynamic Analysis, and Dynamic Simulation. The evolutionary models for populations with competition or intervention are discussed in Evolutionary Models. In Dynamic Analysis, the dynamic behaviors of

populations under different conditions are analyzed. The stability conditions of equilibrium states are also justified for different interventional strategies. In Dynamic Simulation, a two-dimensional dynamic simulation scheme based on the proposed evolutionary model is described. Two sets of simulation results are presented showing potential geographical impacts on the evolution of multilingual populations. To keep it simple and easy to follow, the paper is focused more on bilingual populations, with models for general multilingual populations provided as an appendix in the end. To make the paper more accessible, all formal definitions, theorems, and especially proofs are also organized into the appendix sections. Nonetheless, these appendices are considered to be necessary contents as well for supporting the results and conclusions presented in the paper.

## Evolutionary models

To keep the description simple and easy to follow, consider a bilingual population, the simplest yet most common form of multilingualism. Assume it is large and well mixed, i.e., every individual speaker can interact with all others in the population. If an individual speaker uses two languages $A$ and $B$ with frequencies $x_A$ and $x_B$, respectively, this individual is called an $(x_A, x_B)$-speaker, where $0 \leq x_A, x_B \leq 1$, and $x_A + x_B = 1$. Thus, a $(1, 0)$-speaker is an $A$ speaker, a $(0, 1)$-speaker is a $B$ speaker, and an $(x_A, x_B)$-speaker with $x_A, x_B \neq 0$ is a speaker of both $A$ and $B$, with frequency $x_A$ for $A$ and $x_B$ for $B$.

Likewise, if language $A$ and $B$ are used with frequencies $y_A$ and $y_B$ in average in the whole population, this population is called a $(y_A, y_B)$-population, where $0 \leq y_A, y_B \leq 1$, and $y_A + y_B = 1$. Thus, a $(1, 0)$-population is an $A$ population, a $(0, 1)$-population is a $B$ population, and a $(y_A, y_B)$-population with $y_A, y_B \neq 0$ is a bilingual population of $A$ and $B$, with average frequency $y_A$ for $A$ and $y_B$ for $B$. Here, if the population size is $n$ and speaker $i$ uses $A$ and $B$ with frequencies $x_A^{(i)}$ and $x_B^{(i)}$, respectively, then $y_A = \Sigma_i x_A^{(i)}/n$ and $y_B = \Sigma_i x_B^{(i)}/n$.

### Evolution with competition

Let $P_A(y_A)$ and $P_B(y_B)$ be the payoff functions for $A$ and $B$ speakers in a $(y_A, y_B)$-population, respectively, defined in terms of population percentages and social or economic preferences, with $P_A$ increasing in $y_A$ and $P_B$ in $y_B$, meaning that the larger the population percentage of a language, the more benefit the language provides. Then, the payoff function for a general $(x_A, x_B)$-speaker in a $(y_A, y_B)$-population can be defined in terms of the average use of $A$ and $B$ by this speaker:

$$\pi((x_A, x_B), (y_A, y_B)) = x_A P_A(y_A) + x_B P_B(y_B). \tag{1}$$

For example, the payoff of a $(0.5, 0.5)$-speaker, i.e., a half-$A$ and half-$B$ speaker, in a $(0.5, 0.5)$-population, i.e., a half-$A$ and half-$B$ population, should be $0.5P_A(0.5) + 0.5P_B(0.5)$.

Considering $(x_A, x_B)$ as the strategy of an individual and $(y_A, y_B)$ the strategy of the population, an evolutionary game (Appendix A) can be defined, where every individual tries to maximize his/her payoff. The latter can be achieved when an optimal strategy $(x_A^*, x_B^*)$ is found for every individual. The strategy for the whole population then becomes $(x_A^*, x_B^*)$ as well, and a Nash equilibrium is reached. A Nash equilibrium of this game is thus a strategy $(x_A^*, x_B^*)$ such that

$$\pi((x_A^*, x_B^*), (x_A^*, x_B^*)) \geq \pi((x_A, x_B), (x_A^*, x_B^*))$$
$$\text{for all } (x_A, x_B). \tag{2}$$

Assume for a $(y_A, y_B)$-population that $y_A$ and $y_B$ vary over time as they reach their equilibrium. Then, a system of replicator equations (Appendix A) can also be defined for the population:

$$\begin{cases} \dot{y}_A = y_A y_B (P_A(y_A) - P_B(y_B)) \\ \dot{y}_B = y_B y_A (P_B(y_B) - P_A(y_A)) \end{cases} \tag{3}$$

meaning that the changing rate of average frequency $y_A$ (or $y_B$) of language $A$ (or $B$) depends on the payoff $P_A$ (or $P_B$) compared with the payoff $P_B$ (or $P_A$): if it is higher, $y_A$ (or $y_B$) increases; otherwise, it decreases. Based on evolutionary game theory, a Nash equilibrium of the evolutionary game as conditioned in (2) is always a fixed point of the system of replicator equations given in (3) [18–20] (Appendix A).

## Evolution with intervention

Now consider the situation where some societal influences other than simple competition intervene in the use of languages. Assume that the societal influences or interventions are implemented to counter the arbitrary increase or decrease of either language in a bilingual population. Let $\bar{P}_A(y_A)$ and $\bar{P}_B(y_B)$ be the payoff functions for $A$ and $B$ speakers in a $(y_A, y_B)$-population, respectively, defined in terms of societal influences and interventions, with $\bar{P}_A$ decreasing in $y_A$ and $\bar{P}_B$ in $y_B$, meaning that the smaller the population percentage of a language, the more incentive or less penalty the speakers of the language receive. Here, incentive means more benefit from the society such as more work opportunities or public supports, etc., and penalty means less advantage in the society such as less access to public education or discouragement from the public, etc. Then, the payoff function for a general $(x_A, x_B)$-speaker in a $(y_A, y_B)$-population can be defined in terms of the average use of $A$ and $B$ by this speaker:

$$\bar{\pi}((x_A, x_B), (y_A, y_B)) = x_A \bar{P}_A(y_A) + x_B \bar{P}_B(y_B). \tag{4}$$

It follows that another evolutionary game can be defined, with the Nash equilibrium being a strategy $(x_A^*, x_B^*)$ such that

$$\bar{\pi}((x_A^*, x_B^*), (x_A^*, x_B^*)) \geq \bar{\pi}((x_A, x_B), (x_A^*, x_B^*)) \\ \text{for all } (x_A, x_B), \tag{5}$$

and also a corresponding system of replicator equations:

$$\begin{cases} \dot{y}_A = y_A y_B (\bar{P}_A(y_A) - \bar{P}_B(y_B)) \\ \dot{y}_B = y_B y_A (\bar{P}_B(y_B) - \bar{P}_A(y_A)). \end{cases} \tag{6}$$

Note that the game with competition in (2) has three possible equilibrium strategies. They can be obtained as the fixed points of the system of replicator equations in (3):

$$\begin{aligned} &\text{(a)} \quad y_A^* = 1, \ y_B^* = 0 \\ &\text{(b)} \quad y_A^* = 0, \ y_B^* = 1 \\ &\text{(c)} \quad y_A^*, y_B^* \neq 0, \ P_A(y_A^*) = P_B(y_B^*) \end{aligned} \tag{7}$$

It is not so hard to prove that the strategies in (a) and (b) are evolutionarily stable while the one in (c) is not (see Appendix C). This means that under competition, one of the languages is expected to die eventually while the other takes over the whole population. The co-existence of the two languages is not sustainable.

On the other hand, the game with intervention in (5) also has three possible equilibrium strategies. They can be obtained as the fixed points of the system of replicator equations in (6):

$$
\begin{array}{lll}
\text{(a)} & y_A^* = 1, & y_B^* = 0 \\
\text{(b)} & y_A^* = 0, & y_B^* = 1 \\
\text{(c)} & y_A^*, y_B^* \neq 0, & \bar{P}_A(y_A^*) = \bar{P}_B(y_B^*)
\end{array}
\tag{8}
$$

However in this case, it can be proved that the strategy in (c) is evolutionarily stable while the ones in (a) and (b) are not (see Appendix C). This means that with intervention, it is possible to keep the population of either language not too large or too small. A bilingual population can be maintained.

## Evolution with competition and intervention

Now the two types of games described above can be combined to obtain a complete model for the evolution of a bilingual population. A simple combination can be obtained by merging the systems in (3) and (6) into the following system:

$$
\begin{cases}
\dot{y}_A = y_A y_B [\lambda(P_A(y_A) - P_B(y_B)) + (1 - \lambda)(\bar{P}_A(y_A) - \bar{P}_B(y_B))] \\
\dot{y}_B = y_B y_A [\lambda(P_B(y_B) - P_A(y_A)) + (1 - \lambda)(\bar{P}_B(y_B) - \bar{P}_A(y_A))],
\end{cases}
\tag{9}
$$

where $\lambda \in [0, 1]$ is a scalar. If $\lambda = 1$, the system is reduced to the one in (3) with competition only. If $\lambda = 0$, it is reduced to the one in (6) with intervention only. If $\lambda \in (0, 1)$, the system is in some sense a convex combination of competition and intervention, with more weight on competition when $\lambda > 0.5$ while more on intervention when $\lambda < 0.5$. For convenience, however, a simpler system, with an equal weight given to competition and intervention, will be considered in the following discussion:

$$
\begin{cases}
\dot{y}_A = y_A y_B (P_A(y_A) - P_B(y_B) + \bar{P}_A(y_A) - \bar{P}_B(y_B)) \\
\dot{y}_B = y_B y_A (P_B(y_B) - P_A(y_A) + \bar{P}_B(y_B) - \bar{P}_A(y_A)).
\end{cases}
\tag{10}
$$

In any case, with a combined system, the impacts from language competition and influences of societal intervention are both included for the uses of the languages. The first part of the system, based on language competition, pushes the population to monolingual, while the second part, counted for societal interventions, prevents the population from losing either language. The combination of the two is then expected to provide a more comprehensive account on the changes of the languages and the evolution of the bilingualism.

Note that the system in (10) can be reformulated in a more compact form:

$$
\begin{cases}
\dot{y}_A = y_A y_B (\tilde{P}_A(y_A) - \tilde{P}_B(y_B)) \\
\dot{y}_B = y_B y_A (\tilde{P}_B(y_B) - \tilde{P}_A(y_A)),
\end{cases}
\tag{11}
$$

where

$$
\tilde{P}_A(y_A) = P_A(y_A) + \bar{P}_A(y_A),
\tag{12}
$$

$$
\tilde{P}_B(y_B) = P_B(y_B) + \bar{P}_B(y_B).
\tag{13}
$$

It is then a system of replicator equations with payoff functions $\tilde{P}_A$ and $\tilde{P}_B$, and corresponds to

an evolutionary game, with the Nash equilibrium being a strategy $(x_A^*, x_B^*)$ such that

$$\tilde{\pi}((x_A^*, x_B^*), (x_A^*, x_B^*)) \geq \tilde{\pi}((x_A, x_B), (x_A^*, x_B^*))$$
$$\text{for all } (x_A, x_B), \tag{14}$$

where $\tilde{\pi}$ is the payoff function for the game, and for an $(x_A, x_B)$-speaker in a $(y_A, y_B)$-population

$$\tilde{\pi}((x_A, x_B), (y_A, y_B))$$
$$= \pi((x_A, x_B), (y_A, y_B)) + \bar{\pi}((x_A, x_B), (y_A, y_B)). \tag{15}$$

In general, $P_A$ and $P_B$ can be any increasing functions and $\bar{P}_A$ and $\bar{P}_B$ be any decreasing functions. However, based on previous studies, they can be defined as the following:

$$P_A(y_A) = cy_A^{\alpha-1}s_A, \quad P_B(y_B) = cy_B^{\alpha-1}s_B, \quad 1 < \alpha \leq 2, \tag{16}$$

$$\bar{P}_A(y_A) = \bar{c}y_A^{\bar{\alpha}-1}\bar{s}_A, \quad \bar{P}_B(y_B) = \bar{c}y_B^{\bar{\alpha}-1}\bar{s}_B, \quad 0 \leq \bar{\alpha} < 1, \tag{17}$$

where $c$ and $\bar{c}$ are scaling constants, $\alpha, \bar{\alpha}, s_A, s_B, \bar{s}_A, \bar{s}_B$ are all parameters, $0 \leq s_A, s_B, \bar{s}_A, \bar{s}_B \leq 1$. The parameters $\alpha, \bar{\alpha}$ determine the order of dependency of the payoffs on the population percentages. Since $1 < \alpha \leq 2$, the payoffs from $P_A$ and $P_B$ increase with increasing population percentages. On the other hand, since $0 \leq \bar{\alpha} < 1$, the payoffs from $\bar{P}_A$ and $\bar{P}_B$ decrease with increasing population percentages. The parameters $s_A, s_B$ are used to define the payoffs from competition. They are indicators of social or economic impacts on the payoffs. The larger these values, the more benefit for the corresponding language groups. The parameters $\bar{s}_A, \bar{s}_B$ are used to define the payoffs from interventions. They are rates for language reversing due to interventions. The larger these values, the faster the reversing rates.

The definitions of the functions in (16) and (17) are actually based on previous work on language competition and, in particular, the work by Abrams and Strogatz 2003 [13]. In fact, with these functions, the systems in (3) and (6) are both equivalent to an Abrams-Strogatz system as shown below, differing only in the ranges of the $\alpha$ values and the parameters $c, s_A, s_B$ versus $\bar{c}, \bar{s}_A, \bar{s}_B$:

$$\begin{cases} \dot{y}_A = y_B P(y_A, s_A) - y_A P(y_B, s_B) \\ \dot{y}_B = y_A P(y_B, s_B) - y_B P(y_A, s_A) \end{cases} \tag{18}$$

where $P$ is a function, $P(y, s) = cy^\alpha s$ for some constant $c$. Abrams and Strogatz 2003 surveyed a large number of regions in UK and South America for language competition and estimated the $\alpha$ value around $1.31 \pm 0.25$ [13]. In general, the system can be adopted for modeling language competition for $1 < \alpha \leq 2$ as well as language reversing for $0 \leq \alpha < 1$, as discussed above.

## Dynamic analysis

Continue the assumption of bilingual populations and consider their possible dynamic behaviors based on the models described in previous sections. Assume that the payoffs $P_A, P_B, \bar{P}_A, \bar{P}_B$ are defined by the functions in (16) and (17).

### Dynamics with competition

For competition only, an evolutionary game and a corresponding system of replicator equations are given in (2) and (3). With $P_A$ and $P_B$ given in (16), the game in (2) is actually a so-

called potential game [18, 20] (Appendix B), i.e., there is a potential function $f$ such that

$$\partial f(y_A, y_B)/\partial y_A = P_A(y_A), \quad \partial f(y_A, y_B)/\partial y_B = P_B(y_B). \tag{19}$$

Indeed, $f(y_A, y_B) = (P(y_A, s_A) + P(y_B, s_B))/\alpha$, where $P(y, s) = cy^\alpha s$, $1 < \alpha \leq 2$. Therefore, an equilibrium strategy for the game must be a KKT point of the following potential maximization problem:

$$\max_{y_A, y_B} f(y_A, y_B), \quad y_A + y_B = 1, \quad y_A, y_B \geq 0. \tag{20}$$

Further, if the equilibrium strategy is evolutionarily stable, it must be a strict local maximizer of the potential maximization problem, and vice versa [18, 20] (Appendix B).

For a given strategy $(y_A, y_B)$, the Hessian $H$ of $f$ is given as follows.

$$H(y_A, y_B) = \begin{pmatrix} P'_A(y_A) & 0 \\ 0 & P'_B(y_B) \end{pmatrix} \tag{21}$$

According to the theory on constrained optimization [21, 22], as a KKT point of the potential maximization problem in (20), if an equilibrium strategy for the game in (2) is a local maximizer, the Hessian projected on the null space of the active constraints at this strategy is necessarily negative semidefinite; conversely, if the projected Hessian is negative definite, the equilibrium strategy must be a strict local maximizer of the potential maximization problem [21, 22]. Thus, the evolutionary stability of the equilibrium strategy can be justified by the negative definiteness of the projected Hessian at the strategy (Appendix B).

The game in (2) has three possible equilibrium strategies corresponding to three fixed points of (3), as given in (7). It is easy to see that strategies (a) and (b) are evolutionarily stable (Appendix C). In either case, one language dies while the other takes over the whole population. If $s_A > s_B$, the chance for (a) will be greater than (b), and if $s_B > s_A$, the chance for (b) will be greater than (a). For (c), it is easy to verify that the projected Hessian at this strategy is always positive definite (Appendix C). It follows that the strategy can never be a local maximizer of the potential maximization problem in (20), and can never be evolutionarily stable.

The above behaviors of the game can be further demonstrated in Fig 1, where $P_A$ and $P_B$ are plotted against $y_A$, and the changing directions of $y_A$ are pointed with arrows. The graph in Fig 1(a) shows the behavior of a population with $\alpha = 2$, when $P_A$ and $P_B$ are simple linear functions. The curves of $P_A$ and $P_B$ are displayed over $y_A \in [0, 1]$ with $s_A = 0.75$ and $s_B = 0.25$, when language $A$ dominants $B$. There is an equilibrium strategy $y_A^*$ in $(0, 1)$ such that $P_A(y_A^*) = P_B(y_B^*)$, with $y_A^* = 0.25$ and $y_B^* = 0.75$. It is unstable as $y_A$ decreases to 0 if it starts from below $y_A^*$ while increases to 1 if it starts from above $y_A^*$. Note that $y_A^*$ is relatively small. Therefore, $y_A$ increases to 1 even if $y_A$ starts small.

The graph in Fig 1(b) shows the behavior of a population with $\alpha = 3/2$, when $P_A$ and $P_B$ are square root functions. The curves of $P_A$ and $P_B$ are displayed over $y_A \in [0, 1]$ with $s_A = 0.25$ and $s_B = 0.75$, when language $B$ dominants $A$. There is an equilibrium strategy $y_A^*$ in $(0, 1)$ such that $P_A(y_A^*) = P_B(y_B^*)$, with $y_A^* = 0.9$ and $y_B^* = 0.1$. It is unstable as $y_A$ decreases to 0 if it starts from below $y_A^*$ while increases to 1 if it starts from above $y_A^*$. Note that $y_A^*$ in this case is very large. Therefore, $y_A$ decreases to 0 even if $y_A$ starts large.

## Dynamics with intervention

For intervention only, an evolutionary game and a corresponding system of replicator equations are given in (5) and (6). With $\bar{P}_A$ and $\bar{P}_B$ given in (17), the game in (5) is also a potential

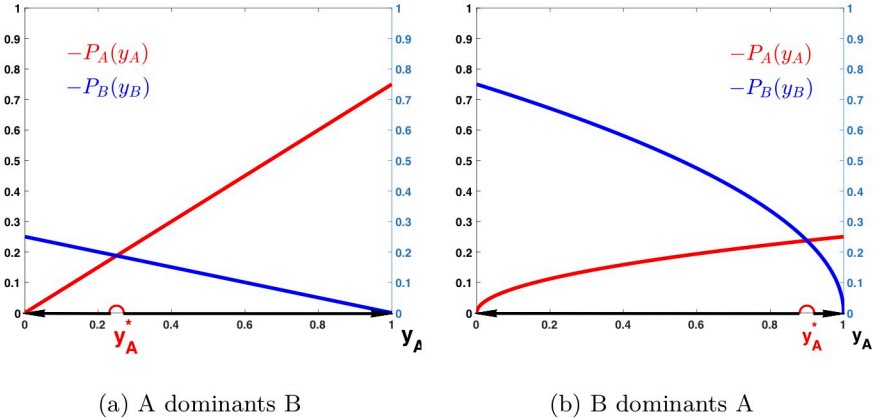

(a) A dominants B (b) B dominants A

**Fig 1. Dynamic behaviors with competition.** Payoff functions $P_A$ and $P_B$ are plotted against $y_A$ and the changing directions of $y_A$ are pointed with arrows. In (a), $\alpha = 2$, $s_A = 0.75$, $s_B = 0.25$, $y_A^* = 0.25$, and $y_B^* = 0.75$. In (b), $\alpha = 3/2$, $s_A = 0.25$, $s_B = 0.75$, $y_A^* = 0.9$, and $y_B^* = 0.1$.

game, i.e., there is a potential function $\bar{f}$ such that

$$\partial \bar{f}(y_A, y_B)/\partial y_A = \bar{P}_A(y_A), \;\; \partial \bar{f}(y_A, y_B)/\partial y_B = \bar{P}_B(y_B). \tag{22}$$

Indeed, for $\bar{\alpha} = 0$, $\bar{f}(y_A, y_B) = \bar{c} \log(y_A)\bar{s}_A + \bar{c} \log(y_B)\bar{s}_B$, and for $0 < \bar{\alpha} < 1$, $\bar{f}(y_A, y_B) = (\bar{P}(y_A, \bar{s}_A) + \bar{P}(y_B, \bar{s}_B))/\bar{\alpha}$, where $\bar{P}(y, \bar{s}) = \bar{c} y \bar{\alpha} \bar{s}$. Therefore, an equilibrium strategy for the game must be a KKT point of the following potential maximization problem:

$$\max_{y_A, y_B} \bar{f}(y_A, y_B), \quad y_A + y_B = 1, \quad y_A, y_B \geq 0. \tag{23}$$

Further, if the equilibrium strategy is evolutionarily stable, it must be a strict local maximizer of the potential maximization problem, and vice versa [18, 20] (Appendix B).

For a given strategy $(y_A, y_B)$, the Hessian $\bar{H}$ of $\bar{f}$ is,

$$\bar{H}(y_A, y_B) = \begin{pmatrix} \bar{P}'_A(y_A) & 0 \\ 0 & \bar{P}'_B(y_B) \end{pmatrix} \tag{24}$$

Again, as a KKT point of the potential maximization problem in (23), if an equilibrium strategy for the game in (5) is a local maximizer, the Hessian projected on the null space of the active constraints at this strategy is necessarily negative semidefinite; conversely, if the projected Hessian is negative definite, the equilibrium strategy must be a strict local maximizer of the potential maximization problem [21, 22]. Thus, the evolutionary stability of the equilibrium strategy can be justified by the negative definiteness of the projected Hessian at the strategy (Appendix B).

The game in (5) has three possible equilibrium strategies corresponding to three fixed points of (6), as given in (8). It is easy to see that strategies (a) and (b) are unstable (Appendix C). For (c), it is easy to verify that the projected Hessian at this strategy is always negative definite (Appendix C). It follows that this strategy must be a strict local maximizer of the potential maximization problem in (23), and must be evolutionarily stable.

The above behaviors of the game can be further demonstrated in Fig 2, where $\bar{P}_A$ and $\bar{P}_B$ are plotted against $y_A$, and the changing directions of $y_A$ are pointed with arrows. The graphs in Fig 2(a) shows the behavior of a population with $\bar{\alpha} = 0$, when $\bar{P}_A$ and $\bar{P}_B$ are reciprocal functions. The curves of $\bar{P}_A$ and $\bar{P}_B$ are displayed over $y_A \in [0, 1]$ with $\bar{s}_A = 0.75$ and $\bar{s}_B = 0.25$,

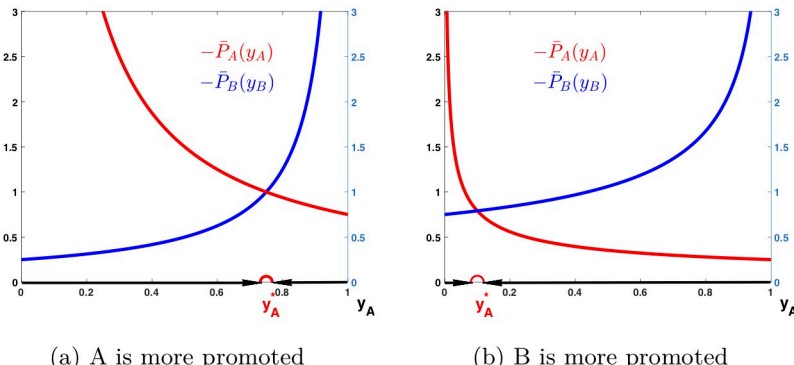

(a) A is more promoted          (b) B is more promoted

**Fig 2. Dynamic behaviors with intervention.** Payoff functions $\bar{P}_A$ and $\bar{P}_B$ are plotted against $y_A$ and the changing directions of $y_A$ are pointed with arrows. In (a), $\bar{\alpha} = 0$, $\bar{s}_A = 0.75$, $\bar{s}_B = 0.25$, $y_A^* = 0.75$, and $y_B^* = 0.25$. In (b), $\bar{\alpha} = 1/2$, $\bar{s}_A = 0.25$, $\bar{s}_B = 0.75$, $y_A^* = 0.1$, and $y_B^* = 0.9$.

when language $A$ is more promoted than $B$. There is an equilibrium strategy $y_A^*$ in $(0, 1)$ such that $\bar{P}_A(y_A^*) = \bar{P}_B(y_B^*)$, with $y_A^* = 0.75$ and $y_B^* = 0.25$. It is stable as $y_A$ increases to $y_A^*$ if it starts from below $y_A^*$ while decreases to $y_A^*$ if it starts from above $y_A^*$.

The graph in Fig 2(b) shows the behavior of a population with $\bar{\alpha} = 1/2$, when $\bar{P}_A$ and $\bar{P}_B$ are inverse square root functions. The curves of $\bar{P}_A$ and $\bar{P}_B$ are displayed over $y_A \in [0, 1]$ with $\bar{s}_A = 0.25$ and $\bar{s}_B = 0.75$, when language $B$ is more promoted than $A$. There is an equilibrium strategy $y_A^*$ in $(0, 1)$ such that $\bar{P}_A(y_A^*) = \bar{P}_B(y_B^*)$, with $y_A^* = 0.1$ and $y_B^* = 0.9$. It is stable as $y_A$ increases to $y_A^*$ if it starts from below $y_A^*$ while decreases to $y_A^*$ if it starts from above $y_A^*$.

## Dynamics with competition and intervention

Now consider the dynamic behaviors of a bilingual population under the influences of both inter-language competition and external interventions, as modeled by the system in (10) and the corresponding game in (14). The payoff functions $\tilde{P}_A$ and $\tilde{P}_B$ are simply combinations of those for competition and intervention as given in (12) and (13). With these two payoff functions, the game in (14) is again a potential game, and therefore, there must be a function $\tilde{f}$ such that

$$\partial \tilde{f}(y_A, y_B)/\partial y_A = \tilde{P}_A(y_A), \quad \partial \tilde{f}(y_A, y_B)/\partial y_B = \tilde{P}_B(y_B). \tag{25}$$

Indeed, $\tilde{f}(y_A, y_B) = f(y_A, y_B) + \bar{f}(y_A, y_B)$, with $f$ and $\bar{f}$ defined in previous sections. It follows that an equilibrium strategy for the game must be a KKT point of the potential maximization problem:

$$\max_{y_A, y_B} \tilde{f}(y_A, y_B), \quad y_A + y_B = 1, \quad y_A, y_B \geq 0. \tag{26}$$

Further, if the equilibrium strategy is evolutionarily stable, it must be a strict local maximizer of the potential maximization problem, and vice versa [18, 20] (Appendix B).

For a given strategy $(y_A, y_B)$, the Hessian $\tilde{H}$ of $\tilde{f}$ is

$$\tilde{H}(y_A, y_B) =$$
$$\begin{pmatrix} P'_A(y_A) & 0 \\ 0 & P'_B(y_B) \end{pmatrix} + \begin{pmatrix} \bar{P}'_A(y_A) & 0 \\ 0 & \bar{P}'_B(y_B) \end{pmatrix}. \tag{27}$$

Again, as discussed in the previous two sections, if an equilibrium strategy for the game in (14) is a local maximizer of the potential maximization problem in (26), the Hessian projected on the null space of the active constraints at this strategy is necessarily negative semidefinite; conversely, if the projected Hessian is negative definite, the equilibrium strategy must be a strict local maximizer of the potential maximization problem [21, 22]. Thus, the evolutionary stability of the equilibrium strategy can be justified by the negative definiteness of the projected Hessian at the strategy (Appendix B).

For convenience, set $c = \bar{c} = 1$ for the following analysis. Let $(y_A^*, y_B^*)$, $y_A^*, y_B^* \neq 0$, be an equilibrium strategy for the evolutionary game in (14). Then, $\tilde{P}_A(y_A^*) = \tilde{P}_B(y_B^*)$, i.e.,

$$P_A(y_A^*) + \bar{P}_A(y_A^*) = P_B(y_B^*) + \bar{P}_B(y_B^*). \tag{28}$$

If $c = \bar{c} = 1$, then

$$(y_A^*)^{\alpha-1} s_A + (y_A^*)^{\bar{\alpha}-1} \bar{s}_A = (y_B^*)^{\alpha-1} s_B + (y_B^*)^{\bar{\alpha}-1} \bar{s}_B. \tag{29}$$

This is a necessary and sufficient condition for $(y_A^*, y_B^*)$, $y_A^*, y_B^* \neq 0$, to be an equilibrium strategy for the evolutionary game in (14). For a specific population, for example, for $\alpha = 3/2$ and $\bar{\alpha} = 0$, it can be simplified to

$$(y_A^*)^{1/2} s_A + (y_A^*)^{-1} \bar{s}_A = (y_B^*)^{1/2} s_B + (y_B^*)^{-1} \bar{s}_B, \tag{30}$$

and for $\alpha = 3/2$ and $\bar{\alpha} = 1/2$, to

$$(y_A^*)^{1/2} s_A + (y_A^*)^{-1/2} \bar{s}_A = (y_B^*)^{1/2} s_B + (y_B^*)^{-1/2} \bar{s}_B. \tag{31}$$

Note that at $(y_A^*, y_B^*)$, $y_A^*, y_B^* \neq 0$, the potential maximization problem in (26) has only one active constraint $y_A + y_B = 1$, and the Hessian projected on the null space of this constraint will be

$$\begin{aligned} \hat{H}(y_A^*, y_B^*) &= (P_A'(y_A^*, y_B^*) + P_B'(y_A^*, y_B^*)) \\ &+ (\bar{P}_A'(y_A^*, y_B^*) + \bar{P}_B'(y_A^*, y_B^*)) \end{aligned} \tag{32}$$

If $c = \bar{c} = 1$, $\hat{H}(y_A^*, y_B^*) < 0$ if and only if

$$\begin{aligned} &(1 - \bar{\alpha})[(y_A^*)^{\bar{\alpha}-2} \bar{s}_A + (y_B^*)^{\bar{\alpha}-2} \bar{s}_B] \\ &> (\alpha - 1)[(y_A^*)^{\alpha-2} s_A + (y_B^*)^{\alpha-2} s_B]. \end{aligned} \tag{33}$$

This is a sufficient condition for an equilibrium strategy $(y_A^*, y_B^*)$, $y_A^*, y_B^* \neq 0$, to be evolutionarily stable. For a specific population, for example, for $\alpha = 3/2$ and $\bar{\alpha} = 0$, the condition can be simplified to:

$$(y_A^*)^{-2} \bar{s}_A + (y_B^*)^{-2} \bar{s}_B > [(y_A^*)^{-1/2} s_A + (y_B^*)^{-1/2} s_B]/2, \tag{34}$$

and for $\alpha = 3/2$ and $\bar{\alpha} = 1/2$, to

$$(y_A^*)^{-3/2} \bar{s}_A + (y_B^*)^{-3/2} \bar{s}_B > (y_A^*)^{-1/2} s_A + (y_B^*)^{-1/2} s_B. \tag{35}$$

An equilibrium strategy $(y_A^*, y_B^*)$, $y_A^*, y_B^* \neq 0$, corresponds to a bilingual population with both language $A$ and $B$ co-existing. In previous sections, competition-only and intervention-only populations have been discussed. For the case with competition-only, such a strategy is never stable, and therefore, the two languages can never co-exist, even in bilingual forms. For the case with intervention-only, such a strategy is always stable, and the two languages can co-exist. By combining the two, it is hoped that such an equilibrium strategy still exists and is also

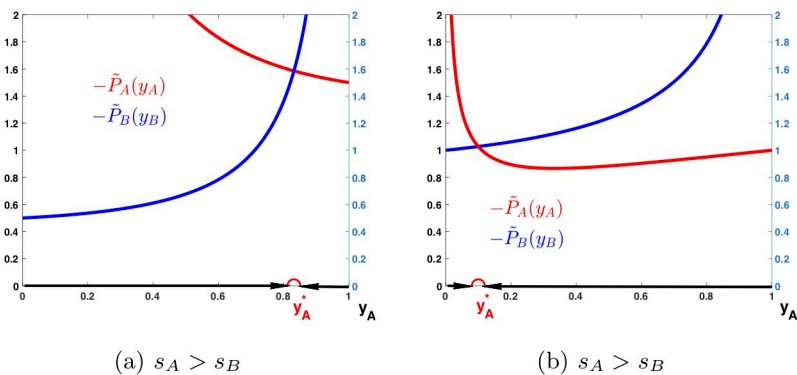

**Fig 3. Dynamic behaviors with competition and intervention.** Payoff functions $\tilde{P}_A$ and $\tilde{P}_B$ are plotted against $y_A$ and the changing directions of $y_A$ are pointed with arrows. In (a), $\alpha = 3/2$ and $\bar{\alpha} = 0$, $\bar{s}_A = s_A = 0.75$ and $\bar{s}_B = s_B = 0.25$, and $y_A^* = 0.8315$ and $y_B^* = 0.1685$. In (b), $\alpha = 3/2$ and $\bar{\alpha} = 1/2$, $\bar{s}_A = s_B = 0.25$ and $\bar{s}_B = s_A = 0.75$, and $y_A^* = 0.1$ and $y_B^* = 0.9$.

stable. Indeed, such an equilibrium can be achieved by choosing appropriate interventional strategies:

**Strategy 1**: If $1 - \bar{\alpha} \geq \alpha - 1$, i.e., $\alpha + \bar{\alpha} \leq 2$, the stability condition in (33) can be satisfied easily by choosing appropriate reversing rates $\bar{s}_A$ and $\bar{s}_B$: Since $\bar{\alpha} - 2 < \alpha - 2$, $(y_A^*)^{\bar{\alpha}-2} > (y_A^*)^{\alpha-2}$ and $(y_B^*)^{\bar{\alpha}-2} > (y_B^*)^{\alpha-2}$, and therefore, the stability condition in (33) is satisfied if $\bar{s}_A$ and $\bar{s}_B$ are sufficiently large, say $\bar{s}_A = ts_A$ and $\bar{s}_B = ts_B$, where $1 \leq t \leq \min\{1/s_A, 1/s_B\}$. With such a choice of $\bar{s}_A$ and $\bar{s}_B$, one can prove that $(y_A^*, y_B^*)$ is also unique (see Appendix D for verification and Fig 3(a) for demonstration).

**Strategy 2**: If $1 - \bar{\alpha} = \alpha - 1$, i.e., $\alpha + \bar{\alpha} = 2$, also set $\bar{s}_A = ts_B$ and $\bar{s}_B = ts_A$ for any $1 \leq t \leq \min\{1/s_A, 1/s_B\}$. Then, $(y_A^*, y_B^*)$, $y_A^*, y_B^* \neq 0$, is an equilibrium strategy for all three type of games, the competition-only game, the intervention-only game, and the combination of the two, i.e., the game with both competition and intervention. Mathematically, $\tilde{P}_A(y_A^*) = \tilde{P}_B(y_B^*)$ if and only if $P_A(y_A^*) = P_B(y_B^*)$ and $\bar{P}_A(y_A^*) = \bar{P}_B(y_B^*)$. In addition, the stability condition in (33) is satisfied for $(y_A^*, y_B^*)$. One can then prove that $(y_A^*, y_B^*)$ is evolutionarily stable and also unique (see Appendix D for verification and Fig 3(b) for demonstration).

**Strategy 3**: In general, given a desired equilibrium strategy $(y_A^*, y_B^*)$, $y_A^*, y_B^* \neq 0$, not necessarily optimal for either the competition-only or intervention-only game, it can be made a true equilibrium strategy for the combined game by choosing appropriate interventional strategies: For $1 - \bar{\alpha} \geq \alpha - 1$, set

$$\bar{s}_A = (y_A^*)^{1-\bar{\alpha}}(y_B^*)^{\alpha-1}s_B, \quad \bar{s}_B = (y_B^*)^{1-\bar{\alpha}}(y_A^*)^{\alpha-1}s_A. \tag{36}$$

Then, $\bar{P}_A(y_A^*) = P_B(y_B^*)$ and $\bar{P}_B(y_B^*) = P_A(y_A^*)$. It follows that $\tilde{P}_A(y_A^*) = \tilde{P}_B(y_B^*)$, and $(y_A^*, y_B^*)$ becomes an equilibrium strategy for the game. In addition, let $y_A^\circ = 1/(1 + (s_A/s_B)^{1/(\alpha-1)})$, and assume that $(y_A^*, y_B^*)$ is selected such that $y_A^* \geq \max\{y_A^\circ, y_B^*\}$ or $y_A^* \leq \min\{y_A^\circ, y_B^*\}$. Then, the condition in (33) is satisfied at $(y_A^*, y_B^*)$, and the strategy is also evolutionarily stable (see Appendix D for verification and Fig 4 for demonstration).

Note that in the last case, when $(\alpha - 1)/(1 - \bar{\alpha})$ is close to 1, the functions $\tilde{P}_A$ and $\tilde{P}_B$ are likely to have more than one intersections over the interval $(0, 1)$ for $y_A$, as demonstrated in the examples in Fig 4. This implies that there may be more than one equilibrium strategies, among which is the desired one. In this case, the interval $(0, 1)$ is divided by the $y_A^*$ values of these strategies, and the $y_A^*$ value of the selected strategy is converged only within a small

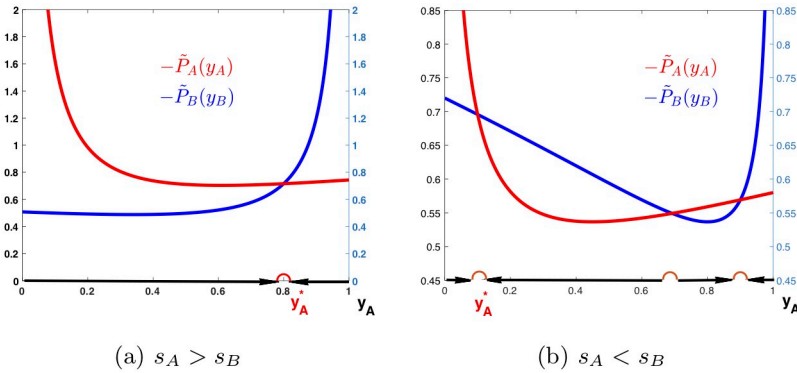

(a) $s_A > s_B$                (b) $s_A < s_B$

**Fig 4. Dynamic behaviors with competition and intervention.** Payoff functions $\tilde{P}_A$ and $\tilde{P}_B$ are plotted against $y_A$ and the changing directions of $y_A$ are pointed with arrows. In (a), $\alpha = 3/2$ and $\bar{\alpha} = 0$, $s_A = 0.6$ and $s_B = 0.4$, $y_A^* = 0.8$ and $y_B^* = 0.2$, and $\bar{s}_A = (y_A^*)(y_B^*)^{1/2} s_B = 0.1431$ and $\bar{s}_B = (y_B^*)(y_A^*)^{1/2} s_A = 0.1073$. In (b), $\alpha = 3/2$ and $\bar{\alpha} = 1/2$, $s_A = 0.4$ and $s_B = 0.6$, $y_A^* = 0.1$ and $y_B^* = 0.9$, and $\bar{s}_A = (y_A^*)^{1/2}(y_B^*)^{1/2} s_B = 0.18$ and $\bar{s}_B = (y_B^*)^{1/2}(y_A^*)^{1/2} s_A = 0.12$.

subinterval around it. When $(\alpha - 1)/(1 - \bar{\alpha})$ is much smaller than 1, however, the functions $\tilde{P}_A$ and $\tilde{P}_B$ are more separated, and usually there is only a single intersection, corresponding to a unique equilibrium strategy.

The dynamic behaviors of the combined game in (14) can be demonstrated through examples shown in Figs 3 and 4. The graph in Fig 3(a) shows the behavior of a population with $\alpha = 3/2$ and $\bar{\alpha} = 0$. The curves of $\tilde{P}_A$ and $\tilde{P}_B$ are displayed over $y_A \in [0, 1]$ with $\bar{s}_A = s_A = 0.75$ and $\bar{s}_B = s_B = 0.25$, when language $A$ has social or economic advantages over $B$. If without intervention, the game would have an equilibrium strategy $(y_A^*, y_B^*)$ with $y_A^* = 0.1$ and $y_B^* = 0.9$, but it is unstable. However, with intervention, the equilibrium strategy becomes $(y_A^*, y_B^*)$ with $y_A^* = 0.8315$ and $y_B^* = 0.1685$. It is stable as $y_A$ increases to $y_A^*$ if it starts from below $y_A^*$ while decreases to $y_A^*$ if it starts from above $y_A^*$.

The graphs in Fig 3(b) shows the behavior of a population with $\alpha = 3/2$ and $\bar{\alpha} = 1/2$. The curves of $\tilde{P}_A$ and $\tilde{P}_B$ are displayed over $y_A \in [0, 1]$ with $\bar{s}_A = s_B = 0.25$ and $\bar{s}_B = s_A = 0.75$. If without intervention, the game would have an equilibrium strategy $(y_A^*, y_B^*)$ with $y_A^* = 0.1$ and $y_B^* = 0.9$, but it is unstable. However, with intervention, the equilibrium strategy remains to be $(y_A^*, y_B^*)$ with $y_A^* = 0.1$ and $y_B^* = 0.9$, but becomes stable as $y_A$ increases to $y_A^*$ if it starts from below $y_A^*$ while decreases to $y_A^*$ if it starts from above $y_A^*$.

Fig 4 shows examples for how to obtain desired equilibrium strategies by choosing appropriate interventional strategies. The graph in Fig 4(a) is an example with $\alpha = 3/2$ and $\bar{\alpha} = 0$. The curves of $\tilde{P}_A$ and $\tilde{P}_B$ are displayed over $y_A \in [0, 1]$ with $s_A = 0.6$ and $s_B = 0.4$, when language $A$ has social or economic advantages over $B$. By setting $\bar{s}_A = (y_A^*)(y_B^*)^{1/2} s_B = 0.1431$ and $\bar{s}_B = (y_B^*)(y_A^*)^{1/2} s_A = 0.1073$, a desired equilibrium strategy $(y_A^*, y_B^*)$ with $y_A^* = 0.8$ and $y_B^* = 0.2$ is obtained. It is stable as $y_A$ increases to $y_A^*$ if it starts from below $y_A^*$ while decreases to $y_A^*$ if it starts from above $y_A^*$.

The graph in Fig 4(b) is an example with $\alpha = 3/2$ and $\bar{\alpha} = 1/2$. The curves of $\tilde{P}_A$ and $\tilde{P}_B$ are displayed over $y_A \in [0, 1]$ with $s_A = 0.4$ and $s_B = 0.6$, when language $B$ has social or economic advantages over $A$. By setting $\bar{s}_A = (y_A^*)^{1/2}(y_B^*)^{1/2} s_B = 0.18$ and $\bar{s}_B = (y_B^*)^{1/2}(y_A^*)^{1/2} s_A = 0.12$, a desired equilibrium strategy $(y_A^*, y_B^*)$ with $y_A^* = 0.1$ and $y_B^* = 0.9$ is obtained. It is stable as $y_A$ increases to $y_A^*$ if it starts from below $y_A^*$ while decreases to $y_A^*$ if it starts from above $y_A^*$. Note that in this example, there are three equilibrium strategies in $(0, 1)$, as marked by

half circles. The desired strategy is one of them. This happens when $(\alpha - 1)/(1 - \bar{\alpha})$ is close or equal to 1.

## Dynamic simulation

Computer simulation can always be used to validate theories when physical experiments are not accessible, and to reveal additional insights into system behaviors which may otherwise be hard to see. This section describes further dynamic behaviors of bilingual populations through computer simulation with the combined model given in (10) and (14). To carry out the simulation, the population is assumed to be distributed in a two-dimentional space, and the changes of the bilingual level of the population will then be tracked across time and space.

More specifically, to carry out the simulation, a 2D torus-shaped lattice of $n \times n$ cells is constructed first, with each cell assumed to be occupied by an individual speaker. A random individual can then be selected repeatedly from the lattice, and a game is played for the individual against the population of the lattice. Let $(x_A, x_B)$ be the current strategy for the individual, and $(y_A, y_B)$ the strategy for the population. Let $p_A = \tilde{P}_A(y_A)$ and $p_B = \tilde{P}_B(y_B)$ be the payoffs for $A$ and $B$ speakers, respectively. Then, the payoff for the individual, $\pi = x_A \tilde{P}_A(y_A) + x_B \tilde{P}_B(y_B)$, is computed. If $p_A > \pi$, $x_A$ is increased by setting $x_A = y_A$ if $x_A < y_A$. On the other hand, if $p_A < \pi$, $x_A$ is reduced by setting $x_A = y_A$ if $x_A > y_A$.

Initially, all individuals are assigned with a random strategy. The game is played $n \times n$ times for the population to complete a generation. The game is repeated for 100 generations to make sure the population reaches its equilibrium. In general, the game can be played in a neighborhood of each selected individual. Let the neighborhood be an $m \times m$ sub-lattice, with the selected individual located at the center. Then, the game can be carried out for each selected individual only against the population in its neighborhood of this size, with the population strategy $(y_A, y_B)$ computed from the population in the neighborhood. Such a game may in fact be more realistic, as people usually interact only with a small group of others around them.

Recall that the population is assumed to be large and well mixed in the game models discussed in previous sections. This means that every individual should be able to interact with all others in the population. With such an assumption, the dynamic behaviors and especially the equilibrium states of the population can be computed by directly solving a corresponding system of replicator equations. The 2D simulation described here can be considered as a discrete solution of the system of replicator equations if the individual in each cell of the lattice is allowed to interact with the individuals in all other cells. However, the 2D simulation is more flexible, allowing restrictions on interactions. It can therefore be used to see how a population behaves when each individual interacts only with a selected group of other individuals, which a continuous model would not be able to reveal.

Fig 5 shows the results from two sets of simulations in two columns of graphs, respectively. The results in the first column are for a population with $\alpha = 2$ and $\bar{\alpha} = 0$, and $\bar{s}_A = s_B = 0.75$ and $\bar{s}_B = s_A = 0.25$. Those in the second column are for a population with $\alpha = 3/2$ and $\bar{\alpha} = 1/2$, and $\bar{s}_A = s_B = 0.4$ and $\bar{s}_B = s_A = 0.6$. In both cases, the population is distributed on a $75 \times 75$ lattice. For each case, the game is played three times with neighborhood size equal to $75 \times 75$, $25 \times 25$, and $5 \times 5$, respectively. The final distribution of the individual frequency $x_A^*$ in the population is displayed for each play in the corresponding order.

In the first column of graphs, the one on the top shows the result from the game with the neighborhood size equal to $75 \times 75$, when each individual interacts with all others in the whole population. The equilibrium frequency $y_A^*$ of the population in this case is approximately equal to 0.75, which agrees with the direct prediction from the continuous model described in previous sections. In addition, the distribution of the individual frequency $x_A^*$ in the population is

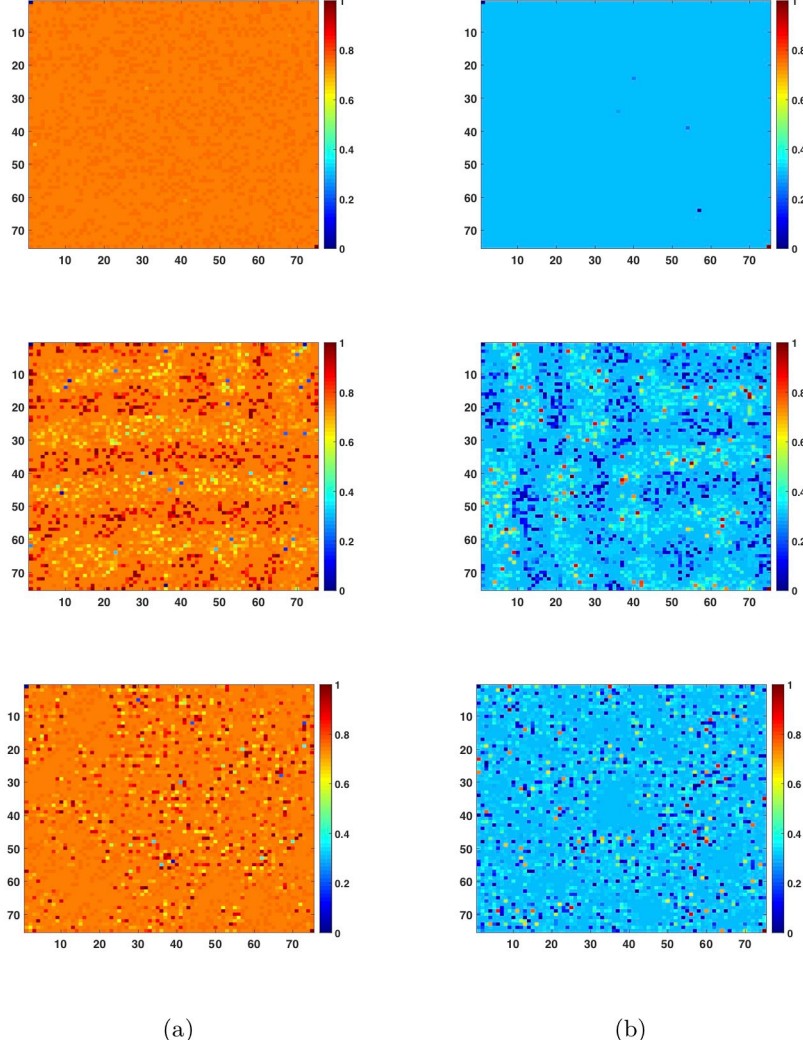

**Fig 5. Dynamic simulation results.** The distributions of the $A$ speaking frequencies in the 2D lattice at equilibrium are displayed in graphs from top to bottom, with corresponding neighborhood sizes equal to $75 \times 75$, $25 \times 25$, and $5 \times 5$. Each of the graphs is a $75 \times 75$ 2D lattice. The $x$-axis and $y$-axis of the graph represent the 75 units of the lattice in the horizontal and vertical directions, respectively. In column (a), $\alpha = 2$ and $\bar{\alpha} = 0$, $\bar{s}_A = s_B = 0.75$ and $\bar{s}_B = s_A = 0.25$, and $y_A^* \approx 0.75$ and $y_B^* \approx 0.25$. In column (b), $\alpha = 3/2$ and $\bar{\alpha} = 1/2$, $\bar{s}_A = s_B = 0.4$ and $\bar{s}_B = s_A = 0.6$, and $y_A^* \approx 0.4$ and $y_B^* \approx 0.6$.

very homogeneous, with $x_A^* \approx 0.75$ across the board, suggesting that language $A$ and $B$ co-exist in the population in an evenly distributed bilingual form. The second graph in this column shows the result from the game with the neighborhood size equal to $25 \times 25$, when the interactions among individual speakers are restricted. The equilibrium frequency $y_A^*$ of the population remains about the same, approximately equal to 0.75. However, the individual frequency $x_A^*$ becomes less constant. Some regions have higher individual frequencies than others, and local groups are formed with varying individual frequencies, as shown in the graph. The graph in the bottom of this column shows the result from the game with the neighborhood size further reduced to $5 \times 5$. While the population frequency $y_A^*$ is not significantly changed, the individual frequency $x_A^*$ shows even bigger variations, with even smaller local spots formed with higher or lower individual frequencies than average.

The graphs in the second column show similar results. Again, when the neighborhood size is set to $75 \times 75$, the equilibrium frequency $y_A^*$ of the population reaches approximately 0.4, which agrees with the direct prediction from the continuous model described in previous sections. The distribution of the individual frequency $x_A^*$ in the population is homogeneous, with $x_A^* \approx 0.4$ across the board, as shown in the graph on the top. However, when the neighborhood size is reduced to $25 \times 25$, the individual frequency $x_A^*$ becomes less constant, although $y_A^*$ remains about the same. Some regions have higher individual frequencies than others, and local groups are formed with varying individual frequencies, as shown in the second graph. When the neighborhood size is further reduced to $5 \times 5$, the individual frequency $x_A^*$ varies in greater degrees in the population. Even smaller local spots are formed with higher or lower individual frequencies than average. The dynamic behaviors shown from these simulations agree with our intuition or experience in language development: Indeed, when communications are restricted to local groups, language variations will remain.

## Discussion

The competition-only model as described in the Evolution with Competition section predicts that between two competing languages, if based only on competition, one would eventually die while the other takes over the whole population, and the co-existing state is unstable. This result is not so surprising, for it has already been discussed in previous studies, although from different perspectives. On the other hand, the intervention-only model as described in the Evolution with Intervention section shows that if controlled only by interventions, it is possible to prevent either language population from becoming too large or too small, and keep the population in a stable co-existing state. By combining the two, the model with both competition and intervention as described in the Evolution with Competition and Intervention section gives a more complete description on the evolution of multilingual populations when it is under the influences of both language competition and societal intervention. It predicts that languages may co-exist stably in multilingual forms if appropriate interventional strategies are employed. In addition, the interventional measures may not only be able to prevent language extinction but also direct populations to desired equilibrium states.

These predictions are based on the assumptions that the payoff functions $P_A$ and $P_B$ are increasing functions and $\bar{P}_A$ and $\bar{P}_B$ are decreasing functions, meaning that in some sense, competition favors large populations, while intervention subsidies small populations. For this reason, the functions in (16) and (17) are adopted for defining $P_A$, $P_B$, $\bar{P}_A$, and $\bar{P}_B$. Besides, $P_A$ and $P_B$ in (16) can be considered as a general set of functions that includes the one used in the Abrams-Strogatz model with $\alpha = 1.31 \pm 0.25$, which is well tested against real language data [13]. The functions for $\bar{P}_A$ and $\bar{P}_B$ in (17) are also related to some applications in genetic selection, where they are used to model genetic mutations with $\bar{\alpha} = 0$ [6, 19]. However, the $\bar{s}_A$ and $\bar{s}_B$ values for genetic mutation are certainly in a much smaller scale than those for language conversion.

Note that this paper has not discussed in detail how the language is acquired, learned, and used, which may in fact relate directly to what language strategies really mean and how they can be changed. There are some obvious questions related to this issue. For example, what does it mean that an individual uses language $A$ with 40% frequency? Is only reading language $A$ without speaking counted as using language $A$? What happens if an individual can never learn a second language? How long does it take for an individual to change his/her language use from one frequency to another? These are legitimate concerns, but may require more linguistic characterizations on language learning and communication, and will hopefully be addressed in future work.

The work in this paper is to develop a general framework for modeling and simulating evolutionary dynamics of multilingual populations. Work remains to be done to analyze some real language groups, while several issues are yet to be resolved: First, some terms used in the proposed models need to be further discussed for exact implications such as "social or economic preferences", which determine the parameters $s_A$ and $s_B$, and "public policies", "educational influences", and "family influences", which the parameters $\bar{s}_A$ and $\bar{s}_B$ depend on. Second, even if exact meanings of these terms are given, the corresponding parameters including $\alpha$ and $\bar{\alpha}$ are not easy to estimate. They may vary with varying populations and varying time periods. The language data is also difficult to collect across a meaningful period of time. Third, the factors that affect language evolution can be more than what a few parameters can cover. While the parameters used in the proposed models are key to determining general dynamic behaviors of multilingual populations, more parameters may be introduced, and additional mechanisms such as the reward and punishment schemes discussed in [23, 24] may be employed.

Computer simulation is a tool to extend the power of theoretical models. It can be used to not only validate the theory but also implement and test additional hypothesis such as the influences from the geographical or demographical structure of the population on multilingual evolution. The simulation done in this study is motivated by previous work on simulation of evolutionary dynamics such as Nowak and May 1992, Durrett and Levin 1997, and more recently, Chen, et al. 2008, Wang, et al. 2012, and Wang, et al. 2016 [25–29]. However, the simulation algorithms can be different for different models and even for the same model. Central to a simulation algorithm is the rule to update the current strategy. Different updating rules may result in very different outcomes. In this work, language competition is considered to be moderate. Therefore, if $p_A > \pi$, i.e., speaking $A$ benefits more, then $x_A$ will stay unchanged if it is already bigger than the average frequency $y_A$. It will increase to $y_A$ only when it is less than $y_A$. The rule could be more aggressive such that $x_A$ will increase by a certain amount even if it is bigger than $y_A$. The results would look different from what have been presented.

Research on language evolution has been pursued extensively in recent years including modeling and simulation. Work has focused on two different but related aspects of language evolution. One is on how a given language develops and evolves with respect to the "genetic changes" of its linguistic elements such as lexicon, syntax, semantics, pronunciations, etc [30–35]. The other is on how a mixed population of multilingual speakers changes dynamically in terms of the percentage of speakers of each language in the whole population [13–17, 36–40]. The work described in this paper belongs to the second category. Work on language death by Abrams and Strogatz 2003 has been followed and extended by several research groups [13–15, 17]. Other approaches have also been proposed, using Lofka-Volterra equations to characterize the evolutionary nature of language dynamics [36, 38], introducing reaction-diffusion models to include geographical influences on language competition [37, 39], and directly simulating the language dynamics with an agent-based or cellular automata models [16, 40]. The model proposed in this work shares some features of these previous approaches, but is focused more on the evolution of multilingualism and in other words, the co-existence of multiple languages in multilingual forms. It is intended to account for possible outside controls or interventions as well as inter-language competitions. Evolutionary games are defined with languages as competing strategies, and corresponding replicator equations are formed to characterize the dynamic behaviors of multilingual populations. The games are also recognized as a special class of potential games and therefore connected to a special class of potential maximization problems, which makes it possible to analyze and even control the equilibrium and stability conditions of given language populations.

To keep the description simple and easy to follow, the proposed models are introduced with and analyzed on bilingual populations only, but they are not limited to bilingual

populations. They can actually be extended straightforwardly to general multilingual populations, as given in Appendix E. In order to make the paper more accessible, efforts have been made to keep the discussions from being too technical, with more formal definitions, theorems, and especially proofs all provided as appendices in the end of the paper. Nonetheless, these appendices are considered to be necessary contents as well for supporting the results and conclusions presented in the paper.

## Appendix A: Evolutionary games [18, 19]

Let $x, y \in S$ be the strategies of an individual and a given population, respectively, where $S = \{x \in R^n : \Sigma_i x_i = 1, x_i \geq 0, i = 1, \ldots, n\}$. Let $\pi(x, y) = \Sigma_i x_i p_i(y)$ be the payoff function for the individual, where $p_i$ is the payoff function for the $i$th pure strategist.

**Definition 1** (Evolutionary Game). *A Nash equilibrium for the evolutionary game with the payoff function $\pi$ is a strategy $x^*$ such that*

$$\pi(x^*, x^*) \geq \pi(x, x^*), \quad \text{for all } x \in S. \tag{37}$$

**Definition 2** (Evolutionary Stability). *An equilibrium strategy $x^*$ is said to be evolutionarily stable if there is a positive number $\bar{\epsilon} < 1$ such that*

$$\pi(x^*, \epsilon x + (1 - \epsilon)x^*) > \pi(x, \epsilon x + (1 - \epsilon)x^*), \tag{38}$$

*for all $x \neq x^*$ and $\epsilon \leq \bar{\epsilon}$,*

**Definition 3** (Replicator Equation). *The following system of equations is called a system of replicator equations*:

$$\dot{x}_i = x_i(p_i(x) - \pi(x, x)), \ \ i = 1, \ldots, n. \tag{39}$$

*where $\pi(x, x) = \Sigma_i x_i p_i(x)$.*

**Theorem 1**. *The equilibrium strategy of an evolutionary game is a fixed point of the system of replicator equations. It is an asymptotically stable fixed point if it is an evolutionarily stable equilibrium strategy.*

**Theorem 2**. *An equilibrium strategy $x^*$ is evolutionarily stable if and only if there is a small neighborhood $N$ of $x^*$ such that $\pi(x^*, x) > \pi(x, x)$ for all $x \in N \cap S, x \neq x^*$.*

## Appendix B: Potential games [18, 20]

**Definition 4** (Potential Game). *A game defined by a payoff function $\pi(x, y) = \Sigma_i x_i p_i(y)$ is called a potential game if there is a function $f$ such that $\partial f(y)/\partial y_i = p_i(y)$ for all $i = 1, \ldots, n$.*

**Definition 5** (Potential Maximization Problem). *The following optimization problem is called a potential maximization problem*:

$$\max_y f(y), \quad y \in S, \tag{40}$$

*where $S = \{y \in R^n : \Sigma_i y_i = 1, y_i \geq 0, i = 1, \ldots, n\}$ and $\partial f(y)/\partial y_i = p_i(y)$ for all $i = 1, \ldots, n$.*

**Theorem 3**. *Let $x^*$ be a strategy for a given potential game. Then, $x^*$ is an equilibrium strategy for the game if and only if it is a KKT point of the corresponding potential maximization problem.*

**Theorem 4**. *Let $x^*$ be an equilibrium strategy for a given potential game. Then, $x^*$ is evolutionarily stable if and only if it is a strict local maximizer of the corresponding potential maximization problem.*

## Appendix C: Equilibrium strategies and stabilities

### Cases for competition only

Consider the game in (2) with the payoff functions $P_A$ and $P_B$ being increasing functions as defined in (16). Let $(y_A^*, y_B^*)$ be an equilibrium strategy in one of the three cases given in (7).

**Theorem 5**. *The equilibrium strategy $(y_A^*, y_B^*)$ is evolutionarily stable for case (a) and (b), but unstable for case (c).*

*Proof.* For case (a), $(y_A^*, y_B^*) = (1, 0)$ and $P_A(y_A^*) > P_B(y_B^*)$. Let $(y_A, y_B)$ be any strategy not equal to $(y_A^*, y_B^*)$. Define $(y_A^\circ, y_B^\circ) = \epsilon(y_A, y_B) + (1 - \epsilon)(y_A^*, y_B^*)$. Then, $\pi((y_A^*, y_B^*), (y_A^\circ, y_B^\circ)) = P_A(y_A^\circ)$. But $\pi((y_A, y_B), (y_A^\circ, y_B^\circ)) = y_A P_A(y_A^\circ) + y_B P_B(y_B^\circ) = P_A(y_A^\circ) + y_B(P_B(y_B^\circ) - P_A(y_A^\circ))$, which is less than $P_A(y_A^\circ)$ for $\epsilon$ sufficiently small. It follows that $(y_A^*, y_B^*)$ must be evolutionarily stable.

For case (b), $(y_A^*, y_B^*) = (0, 1)$ and $P_A(y_A^*) < P_B(y_B^*)$. Let $(y_A, y_B)$ be any strategy not equal to $(y_A^*, y_B^*)$. Define $(y_A^\circ, y_B^\circ) = \epsilon(y_A, y_B) + (1 - \epsilon)(y_A^*, y_B^*)$. Then, $\pi((y_A^*, y_B^*), (y_A^\circ, y_B^\circ)) = P_B(y_B^\circ)$. But $\pi((y_A, y_B), (y_A^\circ, y_B^\circ)) = y_A P_A(y_A^\circ) + y_B P_B(y_B^\circ) = y_A(P_A(y_A^\circ) - P_B(y_B^\circ)) + P_B(y_B^\circ)$, which is less than $P_B(y_B^\circ)$ for $\epsilon$ sufficiently small. It follows that $(y_A^*, y_B^*)$ must be evolutionarily stable.

For case (c), $y_A^* \neq 0$ and $y_B^* \neq 0$. The only active constraint of the problem in (20) at this strategy is $y_A + y_B = 1$. The null space of the constraint at this strategy can be represented by a basis matrix $z = (1, -1)^T$. Then, the projected Hessian on this space at $(y_A^*, y_B^*)$ is $z^T H(y_A^*, y_B^*)z = P_A'(y_A^*) + P_B'(y_B^*)$, which is always positive definite. Therefore, by the second order necessary conditions for constraint optimization, $(y_A^*, y_B^*)$ can never be a local maximizer of the problem in (20), and therefore, can never be evolutionarily stable.

### Cases for intervention only

Consider the game in (5) with the payoff functions $\bar{P}_A$ and $\bar{P}_B$ being decreasing functions as defined in (17). Let $(y_A^*, y_B^*)$ be an equilibrium strategy in one of the three cases given in (8).

**Theorem 6**. *The equilibrium strategy $(y_A^*, y_B^*)$ is evolutionarily unstable for case (a) and (b), but stable for case (c).*

*Proof.* For case (a), $(y_A^*, y_B^*) = (1, 0)$ and $\bar{P}_A(y_A^*) < \bar{P}_B(y_B^*)$. Let $(y_A, y_B)$ be any strategy not equal to $(y_A^*, y_B^*)$. Define $(y_A^\circ, y_B^\circ) = \epsilon(y_A, y_B) + (1 - \epsilon)(y_A^*, y_B^*)$. Then, $\bar{\pi}((y_A^*, y_B^*), (y_A^\circ, y_B^\circ)) = \bar{P}_A(y_A^\circ)$. But $\bar{\pi}((y_A, y_B), (y_A^\circ, y_B^\circ)) = y_A \bar{P}_A(y_A^\circ) + y_B \bar{P}_B(y_B^\circ) = \bar{P}_A(y_A^\circ) + y_B(\bar{P}_B(y_B^\circ) - \bar{P}_A(y_A^\circ))$, which is greater than or equal to $\bar{P}_A(y_A^\circ)$ for $\epsilon$ sufficiently small. It follows that $(y_A^*, y_B^*)$ must be evolutionarily unstable.

For case (b), $(y_A^*, y_B^*) = (0, 1)$ and $\bar{P}_A(y_A^*) > \bar{P}_B(y_B^*)$. Let $(y_A, y_B)$ be any strategy not equal to $(y_A^*, y_B^*)$. Define $(y_A^\circ, y_B^\circ) = \epsilon(y_A, y_B) + (1 - \epsilon)(y_A^*, y_B^*)$. Then, $\bar{\pi}((y_A^*, y_B^*), (y_A^\circ, y_B^\circ)) = \bar{P}_B(y_B^\circ)$. But $\bar{\pi}((y_A, y_B), (y_A^\circ, y_B^\circ)) = y_A \bar{P}_A(y_A^\circ) + y_B \bar{P}_B(y_B^\circ) = y_A(\bar{P}_A(y_A^\circ) - \bar{P}_B(y_B^\circ)) + \bar{P}_B(y_B^\circ)$, which is greater than or equal to $\bar{P}_B(y_B^\circ)$ for $\epsilon$ sufficiently small. It follows that $(y_A^*, y_B^*)$ must be evolutionarily unstable.

For case (c), $y_A^* \neq 0$ and $y_B^* \neq 0$. The only active constraint of the problem in (23) at this strategy is $y_A + y_B = 1$. The null space of the constraint at this strategy can be represented by a basis matrix $z = (1, -1)^T$. Then, the projected Hessian on this space at $(y_A^*, y_B^*)$ is $z^T \bar{H}(y_A^*, y_B^*)z = \bar{P}_A'(y_A^*) + \bar{P}_B'(y_B^*)$, which is always negative definite. Therefore, by the second order sufficient conditions for constraint optimization, $(y_A^*, y_B^*)$ must be a strict local maximizer of the problem in (23), and must therefore be evolutionarily stable.

## Appendix D: Dynamics of interventional strategies

Consider the game in (14) with the payoff functions $\tilde{P}_A$ and $\tilde{P}_B$ defined in (15), where $P_A$ and $P_B$ are given in (16) and $\bar{P}_A$ and $\bar{P}_B$ in (17). Let $(y_A^*, y_B^*), y_A^*, y_B^* \neq 0$, be an equilibrium strategy

or a desired equilibrium strategy for the game. This equilibrium strategy may or may not be stable and unique depending on the choice of interventional strategies $\bar{s}_A$ and $\bar{s}_B$.

**Theorem 7** (Stability and Uniqueness for Strategy 1). *Let* $\bar{s}_A = ts_A$ *and* $\bar{s}_B = ts_B$, $1 \leq t \leq$ min $\{1/s_A, 1/s_B\}$. *Let* $(y_A^*, y_B^*)$, $y_A^*, y_B^* \neq 0$, *be the corresponding equilibrium strategy. Assume that* $1 - \bar{\alpha} \geq \alpha - 1$. *Then,* $(y_A^*, y_B^*)$ *is evolutionarily stable and also unique.*

*Proof.* Since $\bar{\alpha} - 2 < -1 < \alpha - 2$, $(y_A^*)^{\bar{\alpha}-2} > (y_A^*)^{\alpha-2}$ and $(y_B^*)^{\bar{\alpha}-2} > (y_B^*)^{\alpha-2}$. Then, the stability condition in (33) is satisfied at $(y_A^*, y_B^*)$ for all $t$, $1 \leq t \leq \min\{1/s_A, 1/s_B\}$. Therefore, $(y_A^*, y_B^*)$ is evolutionarily stable. Note that the same condition is also satisfied at all strategies $(y_A, y_B)$.

Let $\phi$ be the function for the difference between $\tilde{P}_A$ and $\tilde{P}_B$, $\phi(y_A, y_B) = \tilde{P}_A(y_A) - \tilde{P}_B(y_B)$. Then, $\phi(y_A^*, y_B^*) = 0$. Note that $\phi'_{y_A}(y_A, y_B) = (\alpha - 1)[(y_A)^{\alpha-2}s_A + (y_B)^{\alpha-2}s_B] - (\bar{\alpha} - 1)[(y_A)^{\bar{\alpha}-2}\bar{s}_A + (y_B)^{\bar{\alpha}-2}\bar{s}_B]$. Then, if the condition in (33) is satisfied at all strategies $(y_A, y_B)$, $\phi'_{y_A}(y_A, y_B)$ must be negative, and $\phi(y_A, y_B)$ is monotonically decreasing in $y_A$. It follows that $(y_A^*, y_B^*)$ must be unique.

**Theorem 8** (Stability and Uniqueness for Strategy 2). *Assume that* $1 - \bar{\alpha} = \alpha - 1 = \gamma$. *Let* $\bar{s}_A = ts_B$ *and* $\bar{s}_B = ts_A$, $1 \leq t \leq \min\{1/s_A, 1/s_B\}$. *Let* $(y_A^*, y_B^*)$, $y_A^*, y_B^* \neq 0$, *be the corresponding equilibrium strategy. Then,* $(y_A^*, y_B^*)$ *is unique and also evolutionarily stable.*

*Proof.* Since $(y_A^*, y_B^*)$ is an equilibrium strategy, $\tilde{P}_A(y_A^*) = \tilde{P}_B(y_B^*)$, i.e.,

$$(y_A^*)^\gamma s_A + (y_A^*)^{-\gamma} ts_B = (y_B^*)^\gamma s_B + (y_B^*)^{-\gamma} ts_A. \tag{41}$$

This equation can be written equivalently to

$$(y_A^*)^\gamma s_A + t(y_A^*)^{-\gamma}(y_B^*)^{-\gamma}(y_B^*)^\gamma s_B$$
$$= (y_B^*)^\gamma s_B + t(y_B^*)^{-\gamma}(y_A^*)^{-\gamma}(y_A^*)^\gamma s_A. \tag{42}$$

It can then be re-arranged to

$$((y_A^*)^\gamma s_A - (y_B^*)^\gamma s_B)(1 - t(y_A^*)^{-\gamma}(y_B^*)^{-\gamma}) = 0, \tag{43}$$

which implies that $(y_A^*)^\gamma s_A = (y_B^*)^\gamma s_B$ and hence $(y_A^*)^{-\gamma} s_B = (y_B^*)^{-\gamma} s_A$, i.e., $P_A(y_A^*) = P_B(y_B^*)$ and $\bar{P}_A(y_A^*) = \bar{P}_B(y_B^*)$. Reversely, it is also true.

Since there can only be one strategy $(y_A^*, y_B^*)$ such that $P_A(y_A^*) = P_B(y_B^*)$ and $\bar{P}_A(y_A^*) = \bar{P}_B(y_B^*)$, $(y_A^*, y_B^*)$ is unique.

The uniqueness of $(y_A^*, y_B^*)$ can also be justified by noticing that for any strategy $(y_A, y_B)$, $\phi(y_A, y_B) = \tilde{P}_A(y_A) - \tilde{P}_B(y_B)$ can be written in the following form,

$$\phi(y_A, y_B) =$$
$$(y_A^\gamma s_A - ty_B^{-\gamma}y_A^{-\gamma}y_A^\gamma s_A) - (y_B^\gamma s_B - ty_A^{-\gamma}y_B^{-\gamma}y_B^\gamma s_B), \tag{44}$$

which is equivalent to

$$\phi(y_A, y_B) = (y_A^\gamma s_A - y_B^\gamma s_B)(1 - ty_A^{-\gamma}y_B^{-\gamma}). \tag{45}$$

Note that $\phi(y_A^*, y_B^*) = 0$, $1 - ty_A^{-\gamma}y_B^{-\gamma} < 0$, and $y_A^\gamma s_A - y_B^\gamma s_B$ is negative if $y_A < y_A^*$ and positive if $y_A > y_A^*$. Therefore, $\phi(y_A, y_B)$ is positive if $y_A < y_A^*$ and negative if $y_A > y_A^*$, and $(y_A^*, y_B^*)$ is unique.

For stability, it suffices to show that the stability condition in (33) is satisfied at $(y_A^*, y_B^*)$ for $\bar{s}_A = s_B$ and $\bar{s}_B = s_A$. Note that

$$(y_A^*)^{\bar{\alpha}-2} s_B = (y_A^*)^{\bar{\alpha}-2}(y_B^*)^{1-\alpha}(y_B^*)^{\alpha-1} s_B. \tag{46}$$

Since $(y_B^*)^{\alpha-1} s_B = (y_A^*)^{\alpha-1} s_A$,

$$(y_A^*)^{\bar{\alpha}-2} s_B = (y_A^*)^{\bar{\alpha}-1} (y_B^*)^{1-\alpha} (y_A^*)^{\alpha-2} s_A. \tag{47}$$

It follows that $(y_A^*)^{\bar{\alpha}-2} s_B > (y_A^*)^{\alpha-2} s_A$. Similarly,

$$(y_B^*)^{\bar{\alpha}-2} s_A = (y_B^*)^{\bar{\alpha}-2} (y_A^*)^{1-\alpha} (y_A^*)^{\alpha-1} s_A. \tag{48}$$

Since $(y_A^*)^{\alpha-1} s_A = (y_B^*)^{\alpha-1} s_B$,

$$(y_B^*)^{\bar{\alpha}-2} s_A = (y_B^*)^{\bar{\alpha}-1} (y_A^*)^{1-\alpha} (y_B^*)^{\alpha-2} s_B. \tag{49}$$

It follows that $(y_B^*)^{\bar{\alpha}-2} s_A > (y_B^*)^{\alpha-2} s_B$. The stability condition in (33) is then satisfied and $(y_A^*, y_B^*)$ is evolutionarily stable.

**Theorem 9** (Stability for Strategy 3). *Let* $(y_A^*, y_B^*), y_A^*, y_B^* \neq 0$, *be a given strategy. Let* $\bar{s}_A = (y_A^*)^{1-\bar{\alpha}} (y_B^*)^{\alpha-1} s_B$ *and* $\bar{s}_B = (y_B^*)^{1-\bar{\alpha}} (y_A^*)^{\alpha-1} s_A$. *Assume that* $1 - \bar{\alpha} \geq \alpha - 1$. *Then,* $(y_A^*, y_B^*)$ *is an equilibrium strategy, and is also evolutionarily stable if it is selected such that* $y_A^* > \max\{y_A^\circ, y_B^*\}$ *or* $y_A^* < \min\{y_A^\circ, y_B^*\}$, *where* $y_A^\circ = 1/(1 + (s_A/s_B)^{1/(\alpha-1)})$.

*Proof.* Given $\bar{s}_A = (y_A^*)^{1-\bar{\alpha}} (y_B^*)^{\alpha-1} s_B$ and $\bar{s}_B = (y_B^*)^{1-\bar{\alpha}} (y_A^*)^{\alpha-1} s_A$, it is easy to verify that $\bar{P}_A(y_A^*) = P_B(y_B^*)$ and $\bar{P}_B(y_B^*) = P_A(y_A^*)$, and $\tilde{P}_A(y_A^*) = \tilde{P}_B(y_B^*)$, proving that $(y_A^*, y_B^*)$ is an equilibrium strategy.

Define a function $\psi$ for the difference between the left and right hand sides of the stability condition in (33) without the constant terms $(1 - \bar{\alpha})$ and $(\alpha - 1)$:

$$\psi(y_A, y_B) = (y_A)^{\bar{\alpha}-2} \bar{s}_A + (y_B)^{\bar{\alpha}-2} \bar{s}_B - (y_A)^{\alpha-2} s_A - (y_B)^{\alpha-2} s_B. \tag{50}$$

Let $y_A = y_A^*, y_B = y_B^*$, and substitute $\bar{s}_A$ and $\bar{s}_B$ into the function to obtain:

$$\begin{aligned}\psi(y_A^*, y_B^*) &= (y_A^*)^{-1} (y_B^*)^{\alpha-1} s_B + (y_B^*)^{-1} (y_A^*)^{\alpha-1} s_A \\ &\quad - (y_A^*)^{-1} (y_A^*)^{\alpha-1} s_A - (y_B^*)^{-1} (y_B^*)^{\alpha-1} s_B.\end{aligned} \tag{51}$$

The formula can be further simplified to

$$\begin{aligned}&\psi(y_A^*, y_B^*) \\ &= (y_A^*)^{-1} (y_B^*)^{-1} (y_A^* - y_B^*)((y_A^*)^{\alpha-1} s_A - (y_B^*)^{\alpha-1} s_B) \\ &= (y_A^* y_B^*)^{-1} (y_A^* - y_B^*)(P_A(y_A^*) - P_B(y_B^*)).\end{aligned} \tag{52}$$

Note that $P_A(y_A^\circ) - P_B(y_B^\circ) = 0$ for $y_A^\circ = 1/(1 + (s_A/s_B)^{1/(\alpha-1)})$. Then, since $P_A(y_A) - P_B(y_B)$ is monotonically increasing in $y_A$, $P_A(y_A^*) - P_B(y_B^*)$ is positive if $y_A^* > y_A^\circ$ and negative if $y_A^* < y_A^\circ$. Then, if $(y_A^*, y_B^*)$ is selected such that $y_A^* > \max\{y_A^\circ, y_B^*\}$ or $y_A^* < \min\{y_A^\circ, y_B^*\}$, $\psi(y_A^*, y_B^*) > 0$. Since $(1 - \bar{\alpha}) \geq (\alpha - 1)$, it follows that the stability condition in (33) is satisfied at $(y_A^*, y_B^*)$ and $(y_A^*, y_B^*)$ is evolutionarily stable.

Note that for Strategy 3, the equilibrium strategy $(y_A^*, y_B^*), y_A^*, y_B^* \neq 0$, may not always be unique, especially if $1 - \bar{\alpha}$ is close to $\alpha - 1$. For example, if $1 - \bar{\alpha} = \alpha - 1 = \gamma$, it is easy to verify that $\bar{P}_A(y_B^*) = P_B(y_A^*)$ and $\bar{P}_B(y_A^*) = P_A(y_B^*)$, and $\tilde{P}_A(y_B^*) = \tilde{P}_B(y_A^*)$, which means that $y_A = y_B^*$ and $y_B = y_A^*$ is also an equilibrium strategy of the game, in addition to the equilibrium strategy $y_A = y_A^*$ and $y_B = y_B^*$. It can also be proved evolutionarily stable: For $y_A = y_B^*$ and

$y_B = y_A^*,$

$$\psi(y_B^*, y_A^*) = \\ (y_B^*)^{\bar{\alpha}-2}\bar{s}_A + (y_A^*)^{\bar{\alpha}-2}\bar{s}_B - (y_B^*)^{\alpha-2}s_A - (y_A^*)^{\alpha-2}s_B,$$ (53)

and if $\bar{s}_A = (y_A^*)^{1-\bar{\alpha}}(y_B^*)^{\alpha-1}s_B$ and $\bar{s}_B = (y_B^*)^{1-\bar{\alpha}}(y_A^*)^{\alpha-1}s_A$,

$$\psi(y_B^*, y_A^*) = (y_B^*)^{-1}(y_A^*)^{1-\bar{\alpha}}s_B + (y_A^*)^{-1}(y_B^*)^{1-\bar{\alpha}}s_A \\ -(y_B^*)^{-1}(y_B^*)^{\alpha-1}s_A - (y_A^*)^{-1}(y_A^*)^{\alpha-1}s_B,$$ (54)

which can be simplified to

$$\psi(y_B^*, y_A^*) \\ = (y_A^*)^{-\bar{\alpha}}(y_B^*)^{-\bar{\alpha}}(y_B^* - y_A^*)((y_A^*)^{\bar{\alpha}-1}s_A - (y_B^*)^{\bar{\alpha}-1}s_B) \\ = (y_A^* y_B^*)^{-\bar{\alpha}}(y_B^* - y_A^*)(\bar{P}_A(y_A^*) - \bar{P}_B(y_B^*)).$$ (55)

Note that $\bar{P}_A(y_A^\circ) - \bar{P}_B(y_B^\circ) = 0$ for $y_A^\circ = 1/(1 + (s_B/s_A)^{1/(\bar{\alpha}-1)})$. Then, since $\bar{P}_A(y_A) - \bar{P}_B(y_B)$ is monotonically decreasing in $y_A$, $P_A(y_A^*) - P_B(y_B^*)$ is negative if $y_A^* > y_A^\circ$ and positive if $y_A^* < y_A^\circ$. Then, if $(y_A^*, y_B^*)$ is selected such that $y_A^* > \max\{y_A^\circ, y_B^*\}$ or $y_A^* < \min\{y_A^\circ, y_B^*\}$, $\psi(y_B^*, y_A^*) > 0$. Since $(1 - \bar{\alpha}) \geq (\alpha - 1)$, it follows that the stability condition in (33) is satisfied at $(y_B^*, y_A^*)$ and $(y_B^*, y_A^*)$ is also evolutionarily stable.

## Appendix E: General models for multilingualism

Assume that there is a multilingual population of $m$ languages. Let $x = (x_1, \ldots, x_m)^T$ be the strategy vector for an individual speaker, where $x_i$ is the frequency of the individual to speak language $i$, $\Sigma_i x_i = 1$. Let $y = (y_1, \ldots, y_m)^T$ be the strategy vector for the population, with $y_i$ being the average frequency of the population to speak language $i$, $\Sigma_i y_i = 1$. Thus, an $x$-speaker is an individual speaking language $i$ with frequency $x_i$, and a $y$-population is a population speaking language $i$ with average frequency $y_i$.

Assume that the evolution of the given population is influenced only by competition on population and social or economic preferences. Let $P_i(y_i)$ be the payoff function for the individual who speaks only $i$ when the average frequency of speaking $i$ in the population is given by $y_i$. Assume that $P_i$, $i = 1, \ldots, m$, are monotonically increasing functions. Then, the payoff function $\pi$ for an $x$-speaker in $y$-population can be defined by the average payoff, $\pi(x, y) = \Sigma_i x_i P_i(y_i)$, with which an evolutionary game can be formulated. A Nash equilibrium of this game is a strategy $x^*$ such that $\pi(x^*, x^*) \geq \pi(x, x^*)$ for any strategy $x$. A corresponding system of replicator equations can be defined as

$$\dot{y}_i = \Sigma_j y_i y_j (P_i(y_i) - P_j(y_j)), \quad i = 1, \ldots, m$$ (56)

Similarly, assume that the evolution of the given population is influenced only by possible societal interventions. Let $\bar{P}_i(y_i)$ be the payoff function for the individual who speaks only $i$ when the average frequency of speaking $i$ in the population is given by $y_i$. Assume that $\bar{P}_i$, $i = 1, \ldots, m$, are monotonically decreasing functions. Then, the payoff function $\bar{\pi}$ for an $x$-speaker in $y$-population can be defined by the average payoff, $\bar{\pi}(x, y) = \Sigma_i x_i \bar{P}_i(y_i)$, with which an evolutionary game can be formulated. A Nash equilibrium of this game is a strategy $x^*$ such that $\bar{\pi}(x^*, x^*) \geq \bar{\pi}(x, x^*)$ for any strategy $x$. A corresponding system of replicator equations can be

defined as

$$\dot{y}_i = \Sigma_j y_i y_j (\bar{P}_i(y_i) - \bar{P}_j(y_j)), \quad i = 1, \ldots, m \tag{57}$$

Let $\tilde{P}_i(y_i) = \lambda P_i(y_i) + (1 - \lambda)\bar{P}_i(y_i)$, $i = 1, \ldots, m$, and correspondingly, $\tilde{\pi}(x, y) = \lambda \pi(x, y) + (1 - \lambda)\bar{\pi}(x, y)$, where $\lambda \in [0, 1]$. Then, a general evolutionary game can be formulated for the population with the influences of both language competition and societal intervention. A Nash equilibrium of the game is a strategy $x^*$ such that $\tilde{\pi}(x^*, x^*) \geq \tilde{\pi}(x, x^*)$ for any strategy $x$. A corresponding system of replicator equations can be defined as

$$\dot{y}_i = \Sigma_j y_i y_j (\tilde{P}_i(y_i) - \tilde{P}_j(y_j)), \quad i = 1, \ldots, m. \tag{58}$$

The payoff functions $P_i$ and $\bar{P}_i$ can be defined more specifically such as $P_i(y_i) = cy_i^{\alpha-1}s_i$, $1 < \alpha \leq 2$ and $\bar{P}_i(y_i) = \bar{c}y_i^{\bar{\alpha}-1}\bar{s}_i$, $0 \leq \bar{\alpha} < 1$, where $c$ and $\bar{c}$ are scaling constants, $s_i$ are indicators for social or economic preferences, and $\bar{s}_i$ are language reversing rates due to interventions, $0 \leq s_i, \bar{s}_i \leq 1$.

## Acknowledgments

The author would like to thank the academic editor and the anonymous referees for carefully reading the early version of the manuscript and providing valuable comments and suggestions, and thank Professor Mark Hunacek for his generous help to improve the writing of the manuscript. The author would also like to express special thanks to Dr. Sydney Lamb who once kindly taught the author linguistics and cognitive science during the author's early career, which evidently had a great influence on the author's later interest in language evolution.

All simulation work was done in Matlab. The code is available upon request.

## Author Contributions

**Conceptualization:** Zhijun Wu.

**Data curation:** Zhijun Wu.

**Formal analysis:** Zhijun Wu.

**Funding acquisition:** Zhijun Wu.

**Investigation:** Zhijun Wu.

**Methodology:** Zhijun Wu.

**Project administration:** Zhijun Wu.

**Resources:** Zhijun Wu.

**Software:** Zhijun Wu.

**Supervision:** Zhijun Wu.

**Validation:** Zhijun Wu.

**Visualization:** Zhijun Wu.

**Writing – original draft:** Zhijun Wu.

**Writing – review & editing:** Zhijun Wu.

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
