## [Decision Letter · Decision Letter 0]

17 Aug 2020

PONE-D-20-05223

Why Multilingual, and How to Keep It -- An Evolutionary Dynamics Perspective

PLOS ONE

Dear Dr. Wu,

Thank you for submitting your manuscript to PLOS ONE. After careful consideration, we feel that it has merit but does not fully meet PLOS ONE’s publication criteria as it currently stands. Therefore, we invite you to submit a revised version of the manuscript that addresses the points raised during the review process.

We look forward to receiving your revised manuscript.

Kind regards,

Xiaojie Chen

Academic Editor

PLOS ONE

Journal Requirements:

2. Please consider referring to yourself at the first person of the singular ("I") rather than the plural ("we") in this single-authored paper.

Additional Editor Comments (if provided):

The author should revise the manuscript according to the reviewers' comments.

Reviewers' comments:

Reviewer's Responses to Questions

**Comments to the Author**

1. Is the manuscript technically sound, and do the data support the conclusions?

Reviewer #1: Yes

Reviewer #2: Yes

Reviewer #3: Partly

2. Has the statistical analysis been performed appropriately and rigorously? 

Reviewer #1: Yes

Reviewer #2: Yes

Reviewer #3: N/A

3. Have the authors made all data underlying the findings in their manuscript fully available?

Reviewer #1: Yes

Reviewer #2: Yes

Reviewer #3: Yes

4. Is the manuscript presented in an intelligible fashion and written in standard English?

Reviewer #1: Yes

Reviewer #2: Yes

Reviewer #3: Yes

5. Review Comments to the Author

Reviewer #1: In this MS, the author propose an evolutionary dynamic model to study the evolution of multilingualism. The model consists two different parts. Concretely, the first one relates to selection of languages based on competition and the second one relates to circumstances when selection of languages is altered. Through theoretical analysis and numerical simulation, the results show that the stable co-existence of languages is possible and extinction can be prevented by choosing appropriate interventional strategies. Furthermore, the author gives the stability conditions of equilibrium states for different interventional strategies. Overall, this is a nice and well written paper, tackling an interesting question in the evolution of cooperation. The mathematical analysis is done carefully (although I have to admit that I am unable to check all formulas) and the results are presented in an understandable way.

However, there are some remaining issues with the manuscript, requiring some answers.

1) The author respectively study the evolutionary dynamics in competition-only and intervention-only populations. Furthermore, the author also combines these two and explores the stability of the equilibrium points. He/she make a combination of the systems in (3) and (6) to obtain a new system. Can the author explain in detail why the system can be described by equation 9.

2) In the first part of the manuscript, the author studies the evolutionary dynamics of multilingualism in well-mixed populations, while in the latter part, he/she studies the evolutionary dynamics of multilingual on the 2D torus-shaped lattice. Here, I want to know the relationship between these two, because individuals in the lattice are not well-mixed.

3) The author study how multilingualism may evolve under the language competition and societal interventions. What does societal interventions usually mean here? What is the relationship between this kind of social intervention and some social incentive mechanisms? These social incentive mechanisms include punishment, reward, or exclusion such as Chen et al. PLoS Computational Biology, 14(7), e1006347 (2018), Liu et al. Nonlinear Dynamics, 97(1), 749-766, (2019).

4) The figures in the manuscript should be more exquisite. Concretely, some circles are incomplete. In addition, stable and unstable equilibrium points cannot be distinguished. What does the axis in Figure 5 represent?

5) There are some grammatical problems in the manuscript that need to be further checked. Besides, the format of some references needs to be adjusted, for example, [8], [31-32], [34], [36].

Reviewer #2: The paper investigates the evolution of bilingualism in the framework of evolutionary game theory. The author constructs an evolutionary game-theoretical model between two languages by considering not only the inner competition between them, as Abrams and Strogatz do (Nature 424: 900, 2003), but also the social interventions that are imposed by the external factors such as public policies, education, or family influences. By combining competition with intervention, they show both languages can be stably coexisted in the well-mixed populations. Besides, the author also performs computer simulations to explore the evolutionary dynamics on spatial lattices.

In general, I think this is a good paper. Particularly, it complements the results previously found by Abrams and Strogatz, that is, the evolutionary competition between two languages leads to the extinction of either language. The findings of this paper clearly show us that how bilingualism may evolve under the influences of both competition and intervention.

I didn't check the algebraic manipulations in detail, but the general methods employed seem to be sound and elegant.

I do have some suggestions as to how the style of presentation can be improved (see below). Once these suggestions are incorporated, I support publication.

(1) The title of the manuscript is "Why Multilingual, and How to Keep It -- An Evolutionary Dynamics Perspective". But I find that the author merely investigated the dynamic behaviors of bilingual population. If the author insists on usage of the present title, I think the author should extend the bilingual model to multilingual model, and mainly study the dynamic behaviors of multilingual population.

(2) It seems to me that the author assume that language competition and social intervention have equal impact in the evolution of bilingualism due to the same weights assigned to them in Eq. (9). This maybe a too strong assumption in my opinion. I think it is necessary to explore what happens if they have different impacts in the evolution of bilingualism? Is the present conclusion robust against such changes? If the answer is "No", what are the new results?

(3) With reference to the computer simulations on evolutionary game dynamics in spatial networks, I suggest citing the following papers: Phys. Rev. E 78, 051120 (2008) and Sci. Rep. 2, 740 (2012), wherein the authors used computer simulations for exploring the spatial dynamics of evolutionary games.

Reviewer #3: General evaluation:

My major concern with the paper is as follows: I think it is too theoretical to be published in a general journal such as PLOS ONE. In the current form, I would suggest to submit it to another journal where this kind of article type is common. Moreover, I think that there is a huge gap between empirical data and the model's predictions. I understand that a model with such a high degree of abstraction from reality is hard to be checked against empirical data. However, here even a preliminary attempt is missing (as the author mentioned himself in the Discussion section). Finally, I think there are also some at least questionable aspects of the model definition with respect to how languages are actually acquired, learned or competitively/cooperatively used in real societies. It is not clear to me that the model particularly defines language use. What are the language-specific aspects? It seems to me as if A and B could also be other cultural or social traits and customs, which can be adopted or abandoned, used to initiate competition or cooperation, and supported by societal interventions.

More detailed points:

line 24/25: “It assumes that the speakers are free to choose among available languages.” I think this is a unrealistic assumption. Speakers mostly learn a new language due to their (new) social environment, for example as a result of migration. They are mostly forced to learn the new local language to be able to participate in the social environment. Here it sounds as if speakers can change their language from one day to the other as a free choice.

line 81: “...that the larger the population percentage of a language, the more competitive the speaker of the language.” Why are they more competitive? It there empirical evidence for that? Shouldn't the payoff be dependent either i) on (mutual) intelligibility ii) or (probably more important here) on the social and economic benefits of using a particular language? I am not convinced that majority alone is sufficient for a language to be more competitive per se.

line 107/108: “...meaning that the smaller the population percentage of a language, the more incentive the speaker of the language receives.” What does the payoff actually stands for? Here it represents the incentive a speaker receives. Is this also the case for the competitive scenario? It is not clear at all what the payoff stands for in general. Maybe the author makes this point more clear to the reader.

Again, considering the high degree of abstraction of the model from the real world, it would be good to see how the model predicts real world phenomena. In the introduction, the author points to the many places in the world where multilingualism exists. But is there empirical evidence that interventions are the (unique) reason for its existence/perpetuation? Is it possible to get data about language competition and societal interventions, etc, that the model can be tested against?

Spelling:

Abstract: and playing -> and is playing

Abstract: accounts for selection -> accounts for the selection

In general: the article 'the' is dropped very often: (the) selection, (the) evolution, etc...

line 56: in home -> at home

6. PLOS authors have the option to publish the peer review history of their article (what does this mean?). If published, this will include your full peer review and any attached files.

Reviewer #1: No

Reviewer #2: No

Reviewer #3: No

---

## [Author Response · Author response to Decision Letter 0]

18 Sep 2020

In the submitted files: Response to the Reviewers.

---

## [Decision Letter · Decision Letter 1]

26 Oct 2020

Why multilingual, and how to keep it -- An evolutionary dynamics perspective

PONE-D-20-05223R1

Dear Dr. Wu,

We’re pleased to inform you that your manuscript has been judged scientifically suitable for publication and will be formally accepted for publication once it meets all outstanding technical requirements.

Kind regards,

Xiaojie Chen

Academic Editor

PLOS ONE

Reviewers' comments:

Reviewer's Responses to Questions

**Comments to the Author**

1. If the authors have adequately addressed your comments raised in a previous round of review and you feel that this manuscript is now acceptable for publication, you may indicate that here to bypass the “Comments to the Author” section, enter your conflict of interest statement in the “Confidential to Editor” section, and submit your "Accept" recommendation.

Reviewer #1: All comments have been addressed

Reviewer #2: All comments have been addressed

Reviewer #3: All comments have been addressed

2. Is the manuscript technically sound, and do the data support the conclusions?

Reviewer #1: Yes

Reviewer #2: Yes

Reviewer #3: Yes

3. Has the statistical analysis been performed appropriately and rigorously? 

Reviewer #1: N/A

Reviewer #2: Yes

Reviewer #3: N/A

4. Have the authors made all data underlying the findings in their manuscript fully available?

Reviewer #1: Yes

Reviewer #2: Yes

Reviewer #3: Yes

5. Is the manuscript presented in an intelligible fashion and written in standard English?

Reviewer #1: Yes

Reviewer #2: Yes

Reviewer #3: Yes

6. Review Comments to the Author

Reviewer #1: The author has carefully and thoughtfully revised this paper and it is now suitable for publication.

Reviewer #2: (No Response)

Reviewer #3: One major concern of my former review was the degree of abstraction of the current study and if it fits to a general journal such as PLOS ONE. The author pointed out covincingly that the topic in question - the modeling of multilingual populations - appears to be of high interest for readers in general journals such as PLOS ONE. Moreover, I think that the author treated all my further comments very carefully, which makes the paper much more comprehensible and gives it a much better flow of reading. Finally, the spelling is clearly improved.

7. PLOS authors have the option to publish the peer review history of their article (what does this mean?). If published, this will include your full peer review and any attached files.

Reviewer #1: No

Reviewer #2: No

Reviewer #3: No

---

## [Editor Report · Acceptance letter]

28 Oct 2020

PONE-D-20-05223R1 

Why multilingual, and how to keep it – An evolutionary dynamics perspective 

Dear Dr. Wu:

I'm pleased to inform you that your manuscript has been deemed suitable for publication in PLOS ONE. Congratulations! Your manuscript is now with our production department. 

Kind regards, 

on behalf of

Professor Xiaojie Chen 

Academic Editor

PLOS ONE